

# A GPS water vapor tomography method based on a genetic algorithm

Fei Yang[1,2,3,4], Jiming Guo[1,3,4*], Junbo Shi[1,3*], Xiaoling Meng[2], Yinzhi Zhao[1], Lv Zhou[5] and Di Zhang[1]

[1]School of Geodesy and Geomatics, Wuhan University, Wuhan 430079, China;

[2]Nottingham Geospatial Institute, University of Nottingham, Nottingham NG7 2TU, United Kingdom

[3]Key Laboratory of Precise Engineering and Industry Surveying of National Administration of Surveying, Mapping and Geoinformation, Wuhan University, Wuhan 430079, China;

[4]Research Center for High Accuracy Location Awareness, Wuhan University, China ;

[5]Guilin University of Technology, Guilin 541004, China.

*Correspondence to*: Jiming Guo and Junbo Shi (jmguo@sgg.whu.edu.cn; jbshi@sgg.whu.edu.cn.)

**Abstract.** Water vapor is an important part in the atmosphere, but its spatial and temporal distribution is difficult to detect. Global Positioning System (GPS) water vapor tomography, which can sense three-dimensional water vapor distribution, has been developed as a research point in the fields of GPS meteorology. In this paper, a new water vapor tomography method based on a genetic algorithm (GA) is proposed to overcome the ill-conditioned problem. By using the proposed approach, it

is not necessary to perform the matrix inversion process, and the water vapor tomography is no longer dependent on excessive constraints, priori information and external data, which give rise to many limitations and difficulties. Experiments in Hong Kong under rainy and rainless conditions show a serious ill-conditioned problem in the tomographic matrix by grayscale and condition numbers. Numerical results indicate that the proposed method achieves high levels of agreement and internal/external accuracy with the GAMIT-estimated slant water vapor (SWV) as a reference. Comparative results of water

vapor density (WVD) derived from radiosonde data reveal that the tomographic results based on the GA are in good agreement with that of radiosonde measurements. In comparison to the traditional Least squares method, a reliable tomographic result with high accuracy can be achieved by the GA without the restrictions mentioned-above. Furthermore, the tomographic results in a rainless scenario are better than those of a rainy scenario, and the reasons are discussed in detail.

## 1 Introduction

Water vapor is a major component of the atmosphere and its distribution and dynamics are the main driving force of weather and climate change. A good understanding of water vapor is crucially important for meteorological applications and research such as severe weather forecasting and warnings (Liu et al., 2005). Nevertheless, the variation of water vapor is affected by many factors, including temperature, topography and seasons with rapidly changing characteristics over time, and are prevalent in both the horizontal and vertical directions, which makes it difficult to monitor with high temporal and spatial

resolution (Rocken et al., 1993).



Due to the development of GPS station networks that provide rich data sources containing atmospheric information, it has been considered as a powerful technique to retrieve water vapor. Since Bevis et al. (1992) first envisioned the potential of tomography applying in GPS meteorology, water vapor tomography has become a promising method to improve the spatio-temporal variations of this parameter (Braun et al., 1999; Nilsson et al., 2004; Song et al., 2006; Perler et al., 2011; Rohm,

2012; Dong and Jin, 2018).

In GPS water vapor tomography, the research area should be covered by ground GPS receivers and discretized into a number of cubic closed voxels by latitude, longitude and altitude, each of which has a fixed amount of water vapor at a particular time (Guo et al., 2016). The observations are GPS-derived slant water vapor, the precipitable water in the direction of the signal ray-path, which travels through the troposphere from its top (Zhao and Yao, 2017). After obtaining the precise

measurement of the signal ray distance in each voxel by raytracing its path from receiver to satellite, we can achieve the basic equation for water vapor tomography, which can be expressed in linear form (Flores et al., 2000; Yang et al., 2018):

$$SWV^q = \sum_{i=1}^{n} d_i^q \cdot x_i, \tag{1}$$

where the superscript $q$ is the satellite signal index, $SWV^q$ denotes the $q$th slant water vapor achieved by GPS tropospheric estimation, $n$ is the total number of tomographic voxels discretized. $d_i^q$ denotes the distance of $q$th signal ray inside voxel $i$

which can be obtained by the satellite and station coordinates, and $x_i$ is the water vapor density of voxel $i$. Using all the suitable SWV observations, we can form the tomographic observation equation:

$$y_{m \times 1} = A_{m \times n} \cdot x_{n \times 1}, \tag{2}$$

where $y$ is a column vector of SWV, $m$ is the total number of SWV measurements in tomography, $A$ denotes the intercept matrix containing the distance of the signal ray in each of the voxels, $n$ is the number of voxels in the study area,

and $x$ denotes the vector of the unknown water vapor density.

Since a GPS signal ray can only pass through a small part of the voxels in the study area, the elements of matrix $A$ are likely to be equal to zero, making it a large sparse matrix. In addition, the effective signal rays will concentrate around the zenith due to the unfavourable geometry of the GPS stations and the special structures of the satellites. These all make Eq. (2) ill-conditioned, and it is difficult to obtain the unknowns by performing the inversion of Eq. (2), in the form of $x = A^{-1} \cdot y$.

To circumvent the ill-conditioned problem, many methods are explored within the literature. Flores et al. (2000) added constraints on the vertical, horizontal variability of tomography with additional top constraints to the model so that the matrix inversion can be performed by the singular value decomposition (SVD). However, most of the constraints are based on experience and difficult to match to the actual water vapor distribution, resulting in the deviation of the tomographic results. Bender et al. (2011) utilized an algebraic reconstruction technique (ART), which is an iterative algorithm to solve the

observation equation. Several reconstruction algorithms of the ART family have also been implemented, (e.g. the





multiplicative algebraic reconstruction techniques (MART) and the simultaneous iterations reconstruction technique (SIRT)). The ART techniques are iterative algorithms that proceed observation by observation. It is not necessary to perform the matrix inversion and therefore avoids the ill-conditioned problem. But the tomographic results depend on the exact initial field, the data quality and relaxation parameter (Wang et al., 2014). Nilsson and Gradinarsky (2004) adapted a Kalman filter

approach to estimate tomographic results without adding constraints and performing the inversion. In this method, initializing the filter with an informed estimation of the water vapor field and providing the initial covariance of state equation are based on external data. Some other approaches that enrich the information of the observation equation have been exploited in recent years, such as Constellation Observing System for Meteorology, Ionosphere, and Climate (COSMIC) occultation data by Xia et al. (2013), Interferometric Synthetic Aperture Radar (InSAR) by Benevides et al. (2015), water

vapor radiometer (WVR) and numerical weather prediction by Chen and Liu (2016).

In the above-mentioned tomographic methods, excessive constraints with the matrix inversion, exact priori information or external data are usually used to overcome the ill-conditioned problem. The mandatory usage of excessive constraints will induce limitations in the water vapor tomography, while reliance on the exact priori information will make the tomographic solutions too similar to the priori data and decrease the role of the tomography technique. For external data, it is not possible

to use it in all tomographic experiments. Therefore, this paper proposes a new tomography method based on a genetic algorithm (Section 2). The tomography experiments and results of the analysis are presented in Section 3. Section 4 represents the conclusions.

## 2 Methodology

### 2.1 Troposphere estimation

In water vapor tomography, the observation is slant water vapor which can be converted from slant wet delay (SWD) by the following formula (Adavi and Mashhadi, 2015):

$$SWV = \Pi \times SWD = \frac{10^6}{\rho_w \times \frac{R}{m_w}\left(\frac{k_3}{T_m} + k_2 - \frac{m_w}{m_d} \times k_1\right)} \times SWD, \tag{3}$$

where $\Pi$ denotes conversion factor. $k_1 = 77.604 K \cdot hPa^{-1}$, $k_2 = 70.4 K \cdot hPa^{-1}$, $k_3 = 3.775 \times 10^5 K^2 \cdot hPa^{-1} \cdot \rho_w$ is the liquid water density (unit: g/m³); $R = 8314 Pa \cdot m^3 \cdot K^{-1} \cdot kmol^{-1}$ represents the universal gas constant;

$m_w = 18.02 kg \cdot kmol^{-1}$ and $m_d = 28.96 kg \cdot kmol^{-1}$ indicate the molar mass of water and the dry atmosphere, respectively; $T_m$ denotes the weighted mean temperature which is the ratio of two vertical integrals though the atmosphere (Davis et al., 1985). In practice, an empirical formula is used to achieve approximate $T_m$ by surface temperature $T_s$ in K





$(T_m = 85.63 + 0.668T_s)$ (Liu et al., 2001; Astudillo et al., 2018). And SWD can be obtained as follows (Zhang et al., 2017):

$$SWD = f(ele) \times ZWD + f(ele) \times \cot(ele) \times \left(G_{NS}^w \times \cos(azi) + G_{WE}^w \times \sin(azi)\right) + R, \qquad (4)$$

where $ele$ and $azi$ are the satellite elevation and azimuth, respectively. $f$ denotes the wet mapping function, $G_{NS}^w$ and $G_{WE}^w$ refer to the wet delay gradient parameters in the north-south and east-west direction, respectively. $R$ is the unmodelled atmospheric slant delay, which is included in the zero-differences residuals. ZWD represents zenith wet delay, which is the wet component of zenith total delay (ZTD) affected by water vapor along the satellite signal ray. It can be separated from ZTD by subtracting the zenith hydrostatic delay (ZHD). And ZTD is an average parameter in spatial aspects and can be achieved by GPS observations. As pressure measurements are available at each station, ZHD is calculated by the Saastamoinen model as follows (Saastamoinen, 1972):

$$ZHD = \frac{0.002277 \times P_s}{1 - 0.00266 \times \cos(2\varphi) - 0.00028 \times H}, \qquad (5)$$

where $P_s$ refers to the surface pressure; $\varphi$ and $H$ represent the latitude and the geodetic height of the station, respectively.

## 2.2 Water vapor tomography based on the genetic algorithm

For water vapor tomography based on the genetic algorithm, the first procedure is to construct the tomographic equation. The idea of function optimization is then used to solve the equations (Guo and Hu, 2009; Olinsky et al., 2004), which is similar to the principle of Least squares $V^T PV = \min$ (Flores et al., 2000). Eq. (2) can be converted into the form as follows:

$$\min f(\mathrm{x}) = (y - Ax)^T P (y - Ax), x \in R+, \qquad (6)$$

where the terms are the same as in Eq. (2). In this equation, the values of $x$ that minimize function $f(x)$ are the result of tomography. To achieve the best values of $x$, the traditional method adopts a derivative method which needs matrix inversion in the follow-up. Genetic algorithm, which was first introduced by Holland (1992), provides an adaptive search method to achieve the tomographic results. It is designed to simulate the evolutionary processes in the nature, in which the principle of survival of the fittest is applied to produce better and better approximates to the function. Eq. (6) is regarded as the fitness function that is used to measure the performance of the searched values of $x$ by computing the fitness value (Goldberg, 1989; Venkatesan et al., 2004). Through searching generation after generation, the water vapor result that best fit the function can be found. The specific steps of water vapor tomography based on genetic algorithm are as follows:

1)  Construct the fitness function which is converted from the tomographic equation.





2) Generate some groups representing approximates of $x$ (water vapor density) stochastically, which form the initial population.

3) Select groups from the last generation of the population as parents according to a lower to higher order of the groups of x corresponding to their fitness values.

5  4) Produce offspring groups from parents by crossover and mutation to make up a new set of approximated solution (new generation).

5) Compute the fitness values of the new generation, go back to step 3) and produce the next generation of the population.

6) The search terminates when the number of generations reaches the stopping criteria, a group of approximates meets the requirement of the fitness value, or the calculation time exceeds limitation.

10                          **Table 1.** Parameters of the genetic algorithm

| Parameter | Strategy |
|---|---|
| Population Size | 200 |
| Crossover Fraction | 0.8 |
| Reproduction of Elite Count | 10 |
| Selection Function | Roulette |
| Crossover Function | Intermediate |
| Mutation Function | Adaptive Feasibility |
| Generations of Stopping Criteria | 100*Number of Variables |

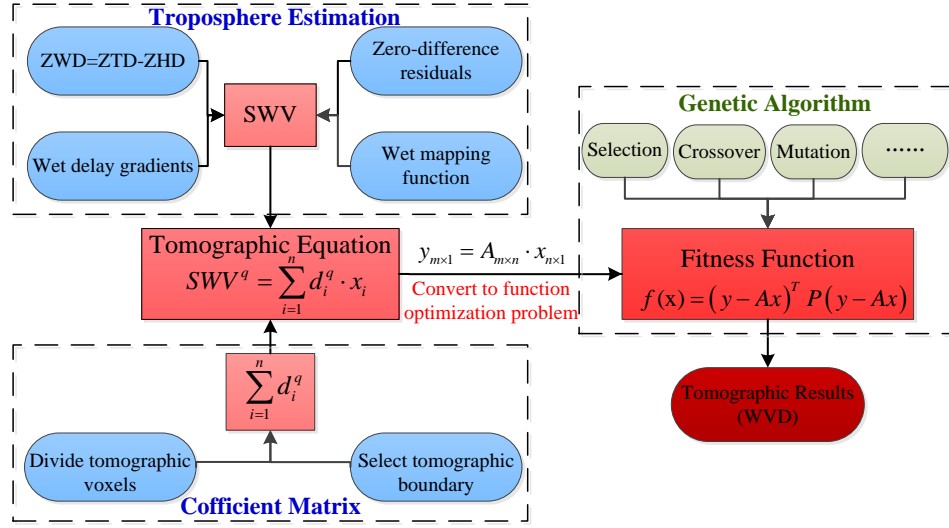

**Fig.1** Flowchart of the water vapor tomography based on the genetic algorithm


The parameters of genetic algorithm are listed in Table 1 (Wang et al., 2010). Roulette is a function used for selection in step 3), referring to the concept of a roulette wheel in which the area of each segment is proportional to its expected value and one of the sections is selected with a random number whose probability equals its area. For the crossover function, Intermediate in Table 1 is intended to create offspring groups by a random weighted average of the parents. The mutation process forces the individuals in the population to undergo small random changes that enable the genetic algorithm to search a wider space. Adaptive feasibility is chosen for the mutation function, which means that the adaptive direction is generated randomly with respect to the last successful or unsuccessful generation (Dwivedi and Dikshit, 2013). Based on these steps, derives the optimal solution of Eq. (6), that is, the value of $x$ that gives $f(x)$ the minimum value, and also the value of water vapor density in the tomographic equations. To more clearly show the process of water vapor tomography based on genetic algorithm, the flowchart is shown in Fig.1.

## 3 Experiment and Analysis

### 3.1 Experiment Description

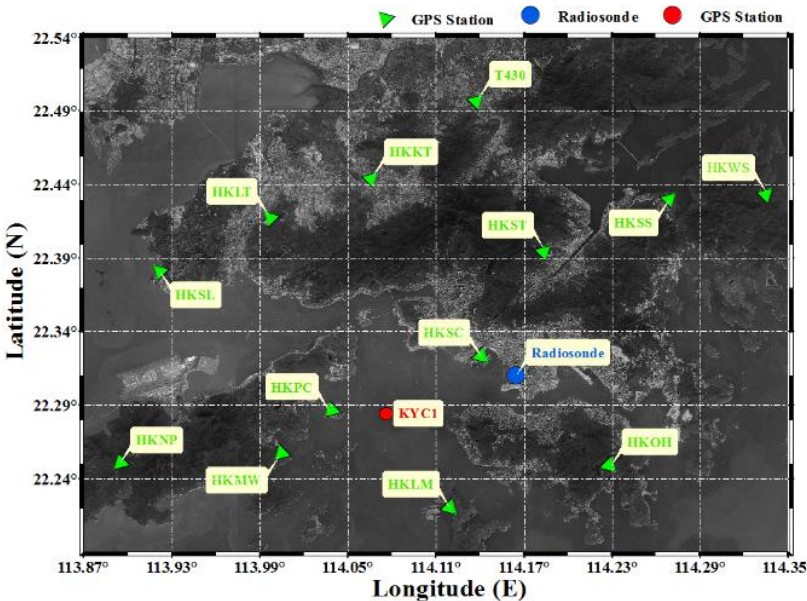

**Fig. 2** Geographic distribution of GPS, radiosonde stations and the horizontal structure of the voxels used in water vapor tomography.

In order to conduct the tomographic experiment based on genetic algorithm, Hong Kong was selected as the research region. The boundary and resolution in west-east and south–north direction were 113.87°-114.35°, 0.06° and 22.19°-22.54°, 0.05°, respectively. For the altitude direction, 0-8.0 km and 800 m were chosen. A total of $8 \times 7 \times 10$ voxels in the tomography grid was obtained. As shown in Figure 2, thirteen GPS stations of the Hong Kong Satellite Positioning Reference Station





Network (green triangle) were selected in the tomography modeling to provide SWV measurements. Another GPS station (KYC1, red spot) and radiosonde station (45005, blue spot) were used to check the result of tomography. Each GPS station recorded temperature, pressure and relative humidity by an automatic meteorological device, by which the hydrostatic parts of the troposphere delay can be accurately achieved. All the stations are under 400 m and located in the first layer of the tomographic grids.

The GPS tropospheric parameters (zenith tropospheric delay and gradient parameters) were estimated by the GAMIT 10.61 software based on a double-differenced model. In order to reduce the strong correlation of tropospheric parameters caused by the short baseline between GPS receivers in the tomographic area, three International GNSS Service (IGS) stations (GJFS, LHAZ and SHAO) were incorporated into the solution model. In the processing, the sampling rate of observations was 30s, a cut-off elevation angle of 10° was selected, and the IGS precise ephemeris was adopted. The LC_AUTCLN and BASELINE were selected as the processing strategies, representing that the GPS observation was the ionosphere-free linear combination and the orbital parameters were fixed, respectively. The tropospheric parameters, including troposphere delay gradients and ZTD at 4-h and 2-h intervals, are estimated and interpolated to 30s sampling rate in the GAMIT software. Note that the outputs of the GAMIT are double-differenced residuals and troposphere delay gradients. To obtain the R in Eq. (4), double-differenced residuals should be converted to zero-differences residuals and multipath effects should be considered by the method proposed by Alber et al. (2000). To achieve the wet delay gradients, Bar-Sever et al. (1998) considered the average of troposphere gradients within 12 hours as the dry delay gradients and subtracted it from the troposphere delay gradients. Then all the necessary parameters were ready for Eq. (4) to build SWD, and SWV was obtained by Eq. (3).

To verify the proposed method, two periods of GPS observation data, with a sampling rate of 30s, were used in the tomography experiment. One from 13 August, 2017 to 19 August, 2017 (DOY of 225 to 231, 2017) during rainless weather. The other is from 12 June, 2017 to 18 June, 2017 (DOY of 163 to 169, 2017) when Hong Kong suffered heavy rain with a maximum daily rainfall of 203.7 mm. The period covered is 0.5h for each tomographic solution. According to the flowchart 1, the results of water vapor density were achieved in every tomographic solution. The radiosonde data, collected twice daily at 00:00 and 12:00 UTC in these two periods, were treated as the reference data.

**3.2 Analysis of matrix ill-condition**

In a tomographic solution, the structure of the coefficient matrix in the observation equation depends on which voxels are crossed by SWV and the number of signal rays penetrating each voxel. Fig. 3 shows it in the form of a grayscale graph. (a) and (b) stand for UTC 00:00 of DOY 225 (a rainless day) and UTC 12:00 of DOY 164 (a rainy day), respectively. In the upper panel of each graph, the deepening of the grayscale refers to an increase in the number of signal rays crossing through the voxel. The closer the layer to the ground, the more voxels are not crossed by any signal rays. Although there are few voxels with no signal rays passing through in the upper layers, many of the voxels have a lighter grayscale which means that the voxels are crossed by fewer signal rays.





Note that when the signal ray passes vertically through the tomographic region, the ray crossed a minimum number of voxels, that is, ten in the tomographic area. Therefore, the minimum probability that a voxel will be crossed by a ray is 1.79% (10/560, 560 is the total number of the voxels in this tomographic experiment). Thus 1.79% of the total SWV is taken as a criteria to further illustrate the structure of the coefficient matrix. If the number is greater than the threshold, the voxel is

5   considered to be crossed by sufficient rays, otherwise the voxel is defined as an insufficient one. For the situation of (a) and (b) (UTC 00:00 of DOY 225 and UTC 12:00 of DOY 164), the number of total SWV and the criteria are 4930/4569 and 88/81, respectively. The lower panel of each graph displays the distribution of sufficient (black rectangle) and insufficient (white rectangle) ones. Obviously, many voxels are not crossed by enough satellite rays, both for the upper layers or the lower layers.

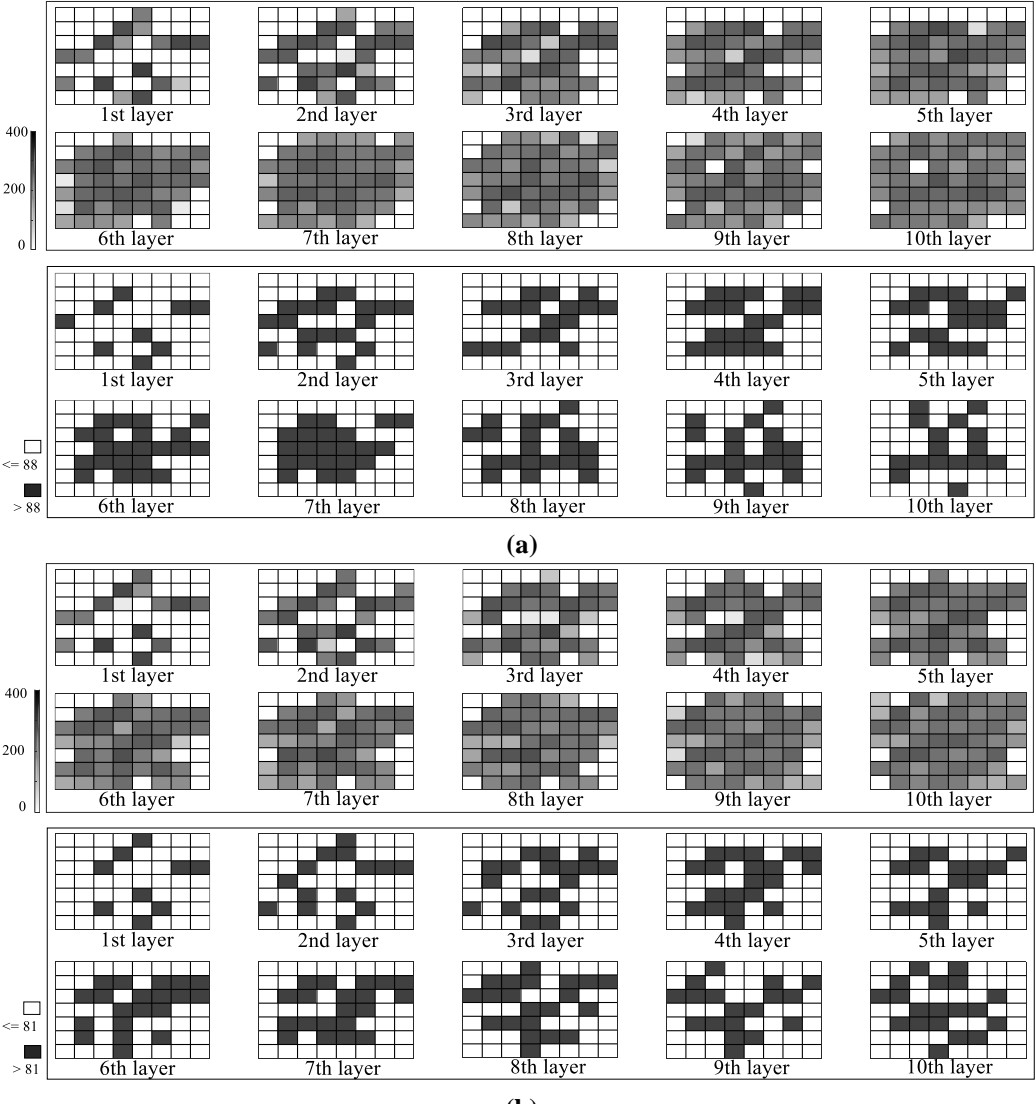



**Fig. 3** Grayscale graph of number of signal rays passing through each voxel and distribution of voxel with sufficient signal rays (a and b stand for a rainless and a rainy day, respectively)

To better analyze the ill-conditioned nature of the observation equation in tomography modeling, the number of zero elements in matrix $A$ is counted. We found that the proportion of zero element is over 97% in all tomographic solutions. In

addition, the concept of matrix condition number is introduced to measure the degree of dispersion of the eigenvalues of the coefficient matrix (Edelman, 1989). The larger the value of the condition number, the more ill-conditioned the matrix is. The results show that the condition number in every tomographic solution is INF which means a serious ill-conditioned problem.

### 3.3 Internal/external Accuracy Testing

To evaluate the performance of water vapor tomography based on genetic algorithm, slant water vapor of GPS stations for

the data of DOY 163 to 169 and DOY 225 to 231, 2017 were computed using the tomographic results based on the water vapor tomographic observation equation established in Eq. (1). In this process, the parameters on the right side of Eq. (1) (the distance of the signal ray in each of the voxels and the water vapor density calculated by the tomographic modelling) are taken as known quantities, and the SWV on the left is the parameter to be determined, called i.e. the tomography-computed SWV. Then the differences against the GAMIT-estimated SWV (as a reference) were also identified.

For internal accuracy testing, 13 GPS stations used in the tomographic modeling were adopted. The change of slant water vapor residuals with elevation angle is shown in Fig. 4, where the blue and red dots represent the rainy and rainless days, respectively. It is clear that the residuals in both weather conditions decreased with ascending elevation angles. The right figure shows that residuals of rainless days are smaller than those of rainy days for all range of elevation angles. The maximum residuals for rainy and rainless scenarios are 10.74 and -9.84 mm, respectively. The root mean square error (RMS)

and mean absolute error (MAE) for rainy and rainless days are 1.56/0.98 and 1.48/0.89mm, respectively. Fig. 4 shows that most of the residuals are concentrated between -2.0 and 2.0 mm, which indicates a good internal accuracy.





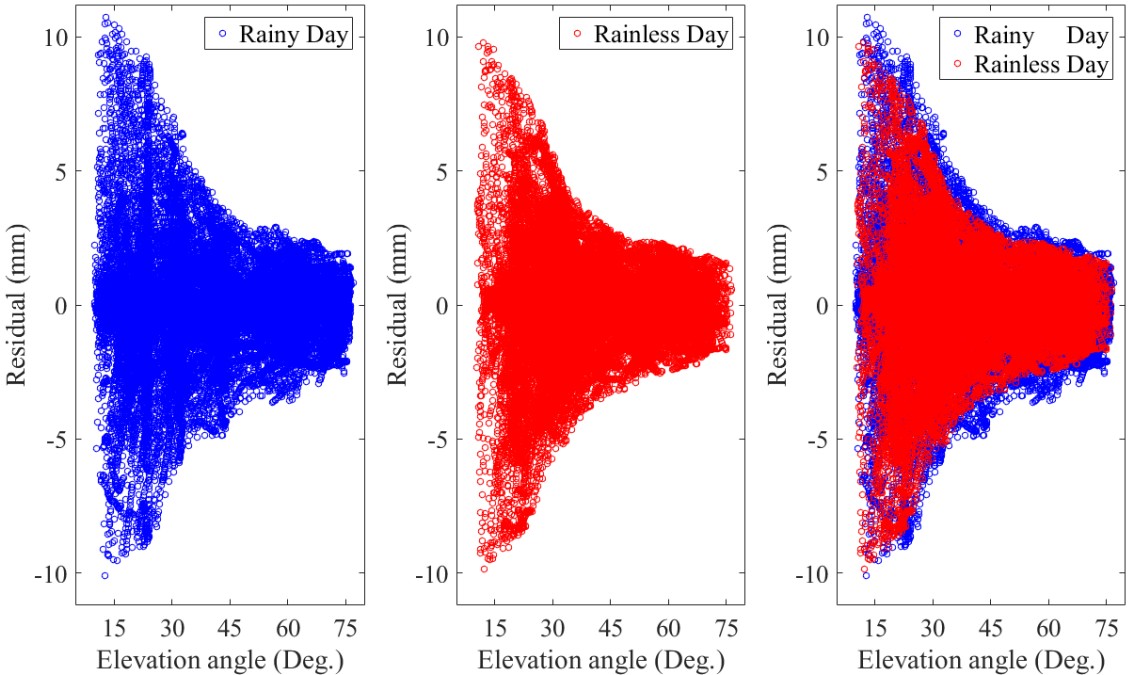

**Fig. 4** Scatter diagram of the SWV residuals in different weather conditions for internal accuracy testing

In order to normalize SWV residuals for their evaluation in a single unit, we mapped the tomography-computed SWVs back to the zenith direction using the $1/\sin(e)$ formula and computed their differences with the GAMIT-estimated PWV (Michal et al., 2017). Fig. 5 shows the statistical results of the residuals in the zenith direction. In the figure, the colours indicate the weather conditions (blue for rainy days and red for rainless days), and the 13 stations were arranged in the order in which they were added to the tomographic model. There were observed mainly as RMS ranging from 0.79 to 1.81 mm, while MAE from 0.43 to 1.54 mm. The RMS and MAE of rainless days are better than those of rainy day in each station. Medians of RMS and MAE are displayed for 13 stations in order to highlight differences among the stations. It is particularly visible for HKMW station where RMS and MAE values over all other stations differ by 1.81/1.53 and 1.60/1.23 for RMS/MAE in rainy and rainless days, respectively. The reason for the divergent behaviour may be that there are two stations (HKPC and HKMW) in the same voxel. This may result in the station (HKPC) data first introduced into the tomographic model affecting the subsequent station (HKMW) data. This specific impact should be discussed in future research. However, plots with RMS and MAE show agreements within 2.0 mm among all the stations (1.5 mm except for HKMW). The residuals statistic in the zenith direction shows a good internal accuracy.





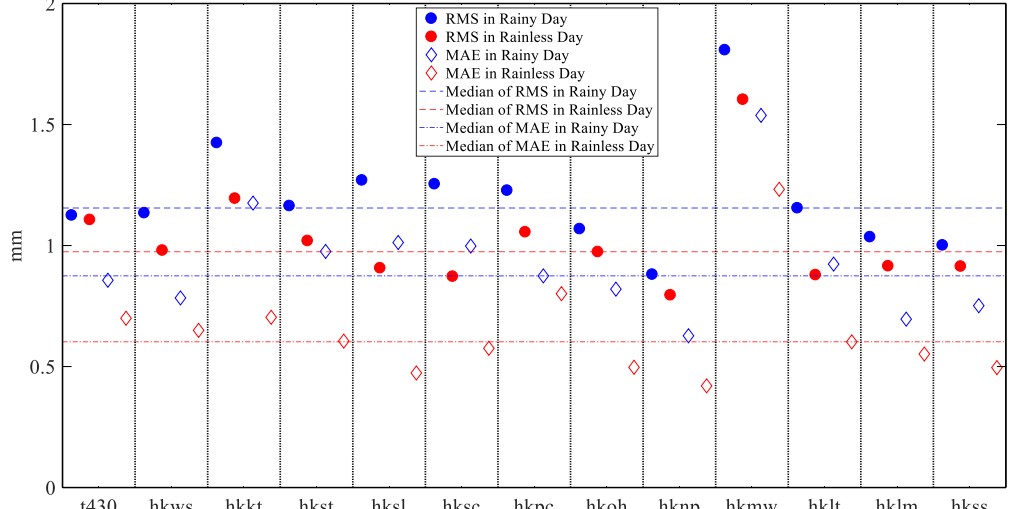

**Fig. 5** Comparison of SWV residuals in zenith direction, circle for RMS and diamond for MAE, blue for rainy days and red for rainless days.

For external accuracy testing, the data from KYC1 station, which was not included in the tomographic modeling, were used.
5 Figure 6 shows the histogram for MAE (upper) and RMS (lower) of SWV residuals, in which the blue ones represent rainy days, reds denote rainless days, and the dashed bars are the averages for different weather conditions. It shows that all the columns in the histogram are below 15 mm, and the reds are generally smaller than the blues, whether in the situation of MEA or RMS. The results of rainless days (8.75/7.33 mm for average RMS/MAE) are better than those of rainy days (11.38/9.54 mm for average RMS/MAE). It is therefore concluded that a good external accuracy is achieved by tomographic
10 solutions considering the low RMS and MAE of rainy and rainless days.



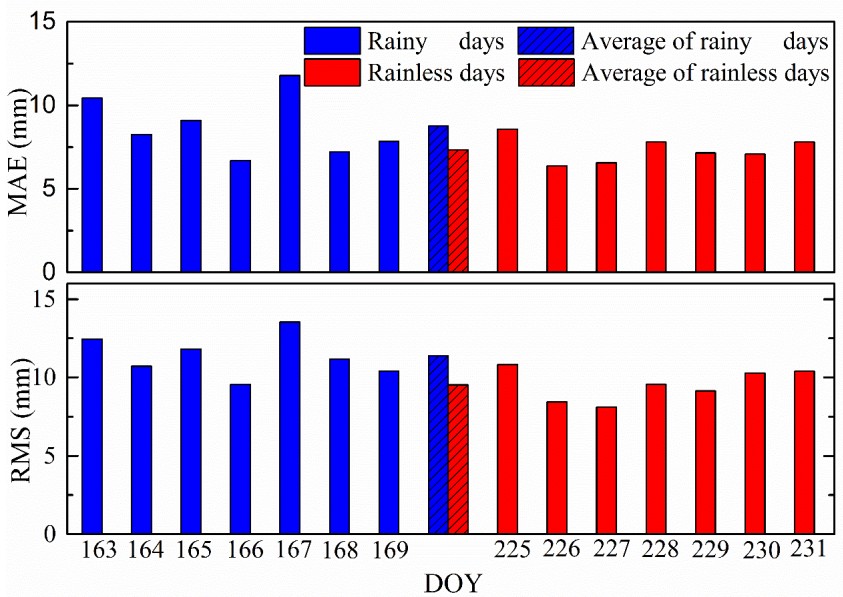

**Fig. 6** Histogram for MAE (upper) and RMS (lower) of SWV residuals in the external accuracy testing (blue for rainy days, red for rainless days)

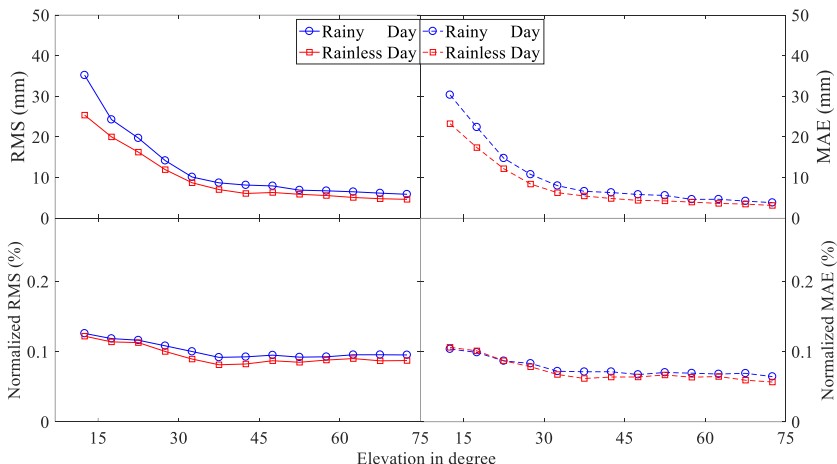

**Fig. 7** Comparison of SWV residuals for each elevation bins, upper for RMS/MAE, lower for normalised RMS/MAE.

To further asses the external accuracy, slant water vapor were grouped into individual elevation bins of 5°, i.e. for example all SWV with an elevation angle between 10° and 15° were evaluated as a single unit. The RMS and MAE of each elevation bin were calculated. In order to examine the dependence of relative errors in SWVs at different elevations, normalized RMS and normalized MAE were computed. For this computation, residuals of SWV were divided by the GAMIT-estimated SWV.

Fig. 7 shows the variation of RMS, MAE, normalized RMS and normalized MAE as the elevation angle changes in different weather conditions. For the upper figures, the RMS and MAE reduction of SWV residuals are clearly visible as the increasing elevation angle, which is consistent with the trend shown in Fig. 4. Colours in the figure indicate that better RMS



and MAE results can be achieved on a rainless day than on a rainy day in each elevation bin. In terms of normalized RMS and MAE, they remain almost constant over all elevation angles, indicating a consistent relative performance of computing SWV among all the weather conditions. It is noted that the normalized RMS and MAE of rainless days are close to those of rainy days. This may be due to the large SWV during rainy days which introduced a larger denominator in the normalized

calculation. Therefore, the good performance on relative error in SWVs at different elevations with a low normalized RMS/MAE (<0.125 for normalized RMS and <0.106 for normalized MAE) demonstrates a good external accuracy.

In the above analysis, RMS and MAE were used for the external accuracy testing of the tomographic results based on the GA. To explore the statistical characteristics of SWV residuals, and to detect the outliers in the tomographic errors, the box plots are used. Five characteristic values are shown in the box plots: Q1 and Q3 located at the bottom and top of the box

represent the first and third quartiles; the second quartile (Q2) is located inside the box; the ends of the whiskers refer to the upper and lower bounds, which are located at Q1-1.5(IQR) and Q3+1.5(IQR), respectively. IQR, that is, the interquartile range, indicating the difference between Q3 and Q1, reflect the discreteness of a set of data. In Fig. 8 the length of box and the range of bound in rainless days (in red) are smaller than those in rainy days (in blue), which illustrates a better residual distribution in rainless days. The right plots (in green) denotes the result of combination of rainless and rainy days,

representing the overall distribution of SWV residuals of tomography based on genetic algorithm. In our experiments, 50 percent of the residuals are concentrated between -7.08 and 4.47 mm, and only 3.24% of the residuals are outliers when combining the data of rainy and rainless days, which show a good distribution of SWV residuals.

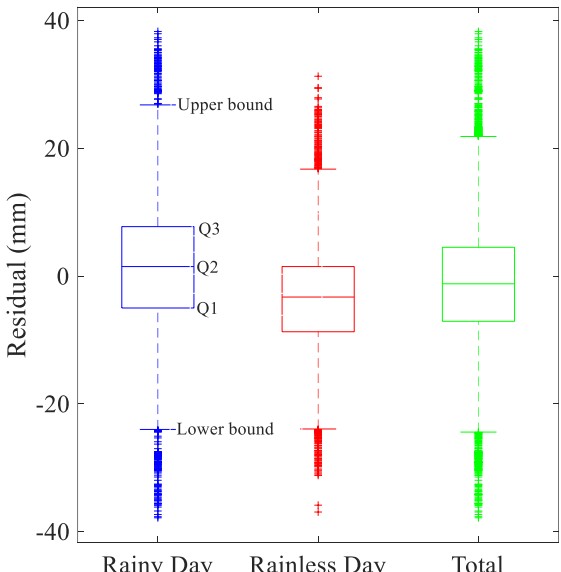

**Fig. 8** Box plots of the SWV residuals



## 3.4 Comparison with Radiosonde Data

**Fig 9 (a-n)** represent water vapor density comparisons between radiosonde and tomography based on the genetic algorithm at UTC 0:00 and 12:00 from DOY 163 to 169, 2017 (rainy days)

5 The water vapor density profile derived from the radiosonde data can be used as a reference value, which is well suited to evaluate the accuracy of the tomographic results based on genetic algorithm. Since the radiosonde launches at UTC 00:00 and 12:00 daily, the tomographic results of DOY 163 to 169 (rainy days) and 225 to 231, 2017 (rainless days) at these time points were compared. Figure 9 shows the water vapor density comparisons between radiosonde data and tomographic





results for different altitudes at individual dates (rainy period). It is clear that all profiles (red lines) and the scatter of radiosonde data (blue dots) decrease with increasing height. The WVD profiles reconstructed by the GA tomographic solutions are in conformity with those derived from the radiosonde data, especially in the upper troposphere. The reason for this phenomenon is that the value of water vapor in the upper layers is relatively low, while water vapor content accounts for

5   more than 90% below 5km near the Earth's surface. In some cases, a relatively good consistency can also be seen in the lower atmosphere. This may be because there is a GPS station (HKSC) for tomography modeling located in the voxel where the radiosonde station is situated, resulting in the lower atmosphere with a sufficient signal rays passing through.

**Table 2** Statistical results of the water vapor density comparison between radiosonde and tomography based on the genetic algorithm for different weather conditions (g/m$^3$)

| Weather condition | DOY | RMS | | MAE | |
|---|---|---|---|---|---|
| | | UTC 0:00 | UTC 12:00 | UTC 0:00 | UTC 12:00 |
| **Rainy days** | 163 | 1.54 | 1.68 | 1.27 | 1.43 |
| | 164 | 1.20 | 1.57 | 1.81 | 1.39 |
| | 165 | 1.37 | 1.79 | 0.85 | 1.56 |
| | 166 | 1.63 | 1.38 | 1.41 | 1.27 |
| | 167 | 1.77 | 1.48 | 1.56 | 1.31 |
| | 168 | 1.49 | 1.33 | 1.55 | 1.18 |
| | 169 | 1.52 | 1.38 | 1.34 | 1.22 |
| | Average | 1.51 | | 1.29 | |
| **Rainless days** | 225 | 1.44 | 1.35 | 1.14 | 0.93 |
| | 226 | 1.46 | 1.25 | 1.18 | 1.05 |
| | 227 | 1.54 | 1.27 | 1.26 | 0.83 |
| | 228 | 1.29 | 1.14 | 1.03 | 0.89 |
| | 229 | 1.38 | 1.39 | 1.09 | 1.24 |
| | 230 | 1.46 | 1.26 | 1.19 | 1.06 |
| | 231 | 1.23 | 1.40 | 1.03 | 1.19 |
| | Average | 1.35 | | 1.08 | |
| **Total** | | 1.43 | | 1.19 | |

10   To further illustrate the comparison with the radiosonde data, Table 2 listed the statistical results of WVD (RMS and MAE). In the table, the WVD in the voxels above the radiosonde station computed by tomography and those derived from radiosonde are counted to calculate their RMS and MAE in each solution. This shows that the average RMS/MAE of rainless days are 1.35/1.08 g/m$^3$, which is smaller than 1.51/1.29 g/m$^3$ in rainy days. It is consistent with the comparison of SWV above. Taking into account the WVD comparison results of Hong Kong tomographic experiments conducted by other





researchers, for example, Xia et al. (2013) obtained a RMS of 1.01 g/m³ by adding the COSMIC profiles, Yao et al. (2016) obtained a RMS of 1.23 g/m³ by maximally using GPS observations and a RMS of 1.60 g/m³ without the operation, Zhao et al. (2017) achieved a RMS of 1.19 g/m³ and 1.61 g/m³ considering the signal rays crossing from the side of the research area and a RMS of 1.79 g/m³ without this consideration, Ding et al. (2017) obtained a RMS of 1.23 g/m³ and 1.45 g/m³ by

5 utilizing the new parametric methods and the traditional methods, Yao et al. (2017) achieved the RMS from 1.48-1.80 g/m³ using different voxel division approaches, etc, the total RMS of 1.43 g/m³ for the two time periods in this paper can be considered as a good agreement with the radiosonde data regardless of the weather conditions. Moreover, it should also be noted that there are many different settings in tomographic experiments by different groups, such as the selection of tomographic boundary, differences of experimental period and weather condition, division rule of horizontal and vertical

voxel, addition of other observations.

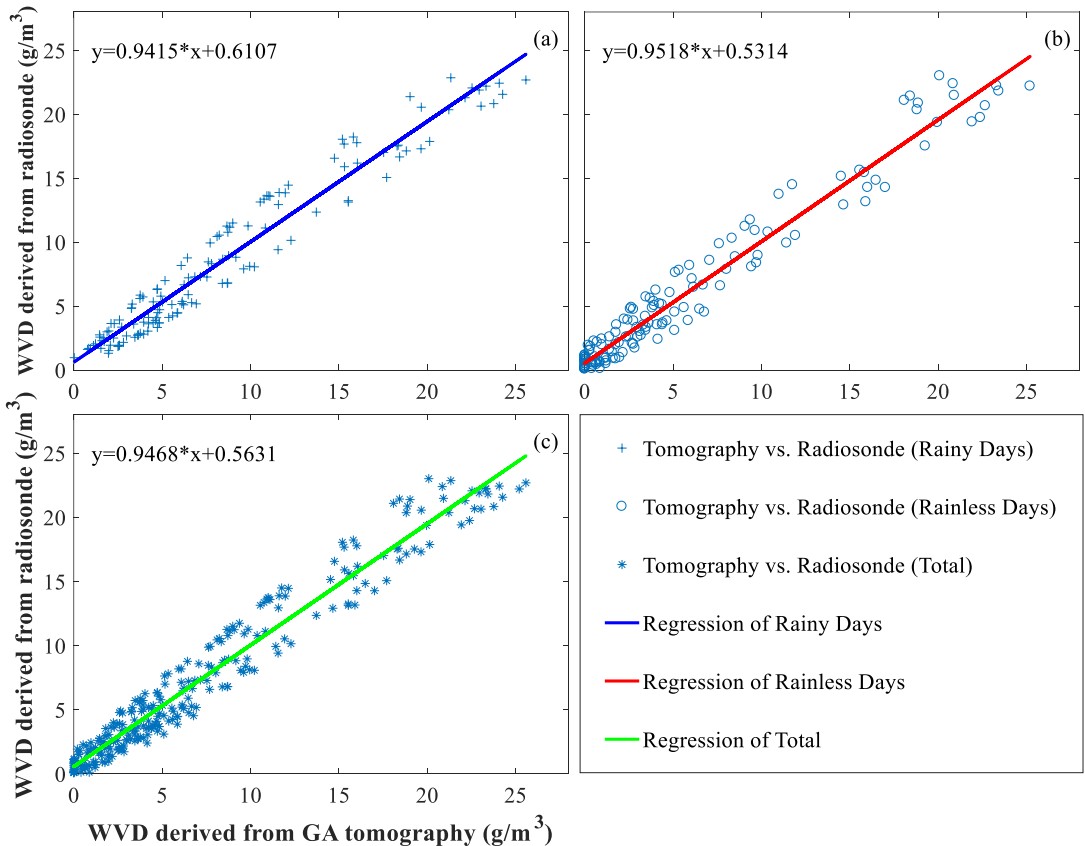

**Fig. 10** Linear regression of the water vapor density from radiosonde and tomography based on the genetic algorithm. (a), (b) and (c) represent rainy days, rainless days and their combination, respectively.

To explore the overall accuracy of water vapor density reconstructed by the GA tomography, the linear regression analysis

and box plot were adopted for different weather conditions. Figure 10 shows the linear regression of the water vapor density for rainy days (a), rainless days (b) and their combination (c), in which the scatter points of three graphs are close to the 1:1





lines. Compared with the coefficients of regression equations, the results from rainless days are slightly better than those of rainy days. When combining the data of two periods, the starting point of the regression equation is 0.5631 and the slope is 0.9468, which indicates that water vapor density with high accuracy can be achieved by tomography based on the GA. Figure 11 shows the box plots, in which the WVD residuals are concentrated in the range of -2 to 2 mm, and the rainless scenario is better than the rainy scenario. The Q1/Q3 are -1.28/1.08, -1.20/0.65 and -1.24/0.87 mm for rainy days, rainless days and their combination, respectively. The upper and lower boundaries are located near 4 mm and -4mm. There are no outliers present in this box plots probably due to the small number of WVD residuals.

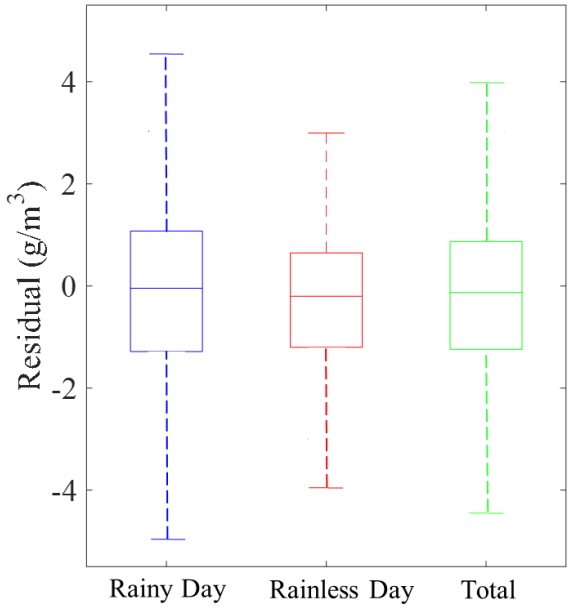

**Fig. 11** Box plots of the WVD residuals

### 3.5 Comparison with tomographic results of the Least Squares method

The Least squares method is most commonly used in water vapor tomography, and it has been proven by a large number of experiments that water vapor density with high accuracy can be obtained with this method (Flores, et al., 2000; Zhang et al., 2017; Zhao et at., 2017). To verify the accuracy of the genetic algorithm, we compared the tomographic results between the genetic algorithm and the least squares method in this section. The specific process and introductions of the least squares method can be seen in detail in previous articles (Flores et al., 2000, Guo et al., 2016, Yang et al., 2018). Figure 12 shows the three-dimensional distribution of water vapor density derived from tomography based on the GA and the least squares method. The water vapor computed by the European Centre for Medium-Range Weather Forecasts (ECMWF) data, which provides various meteorological parameters according to the pressure layer with a spatial resolution of 0.125°*0.125°, is displayed in the figure as a reference. Here both the GA and the least squares method can obtain a reasonable tomographic result. For tomographic results in some voxels, the GA achieves the closer results to the ECMWF data, while for other



voxels, the least squares method performs better. In general, both methods (the GA and the least squares) have a good consistency with ECMWF data regardless of the weather conditions, and can accurately describe the spatial distribution of water vapor. Additionally, a larger variation of water vapor with altitude occurs in a rainy scenario than in a rainless scenario, especially in the upper atmosphere, which is well captured by the GA and the least squares method. Numerical results
5     including RMS and MAE during the whole experimental period are listed in Table 3 to show the comparison of the GA and the least squares method, in which the water vapor density derived from ECMWF data is regarded as the true value. It indicates that the result of the GA is a little better than that of ;east squares method. Actually, the GA is not superior to the least squares method in all solutions, the least squares method yields better results in some solutions.

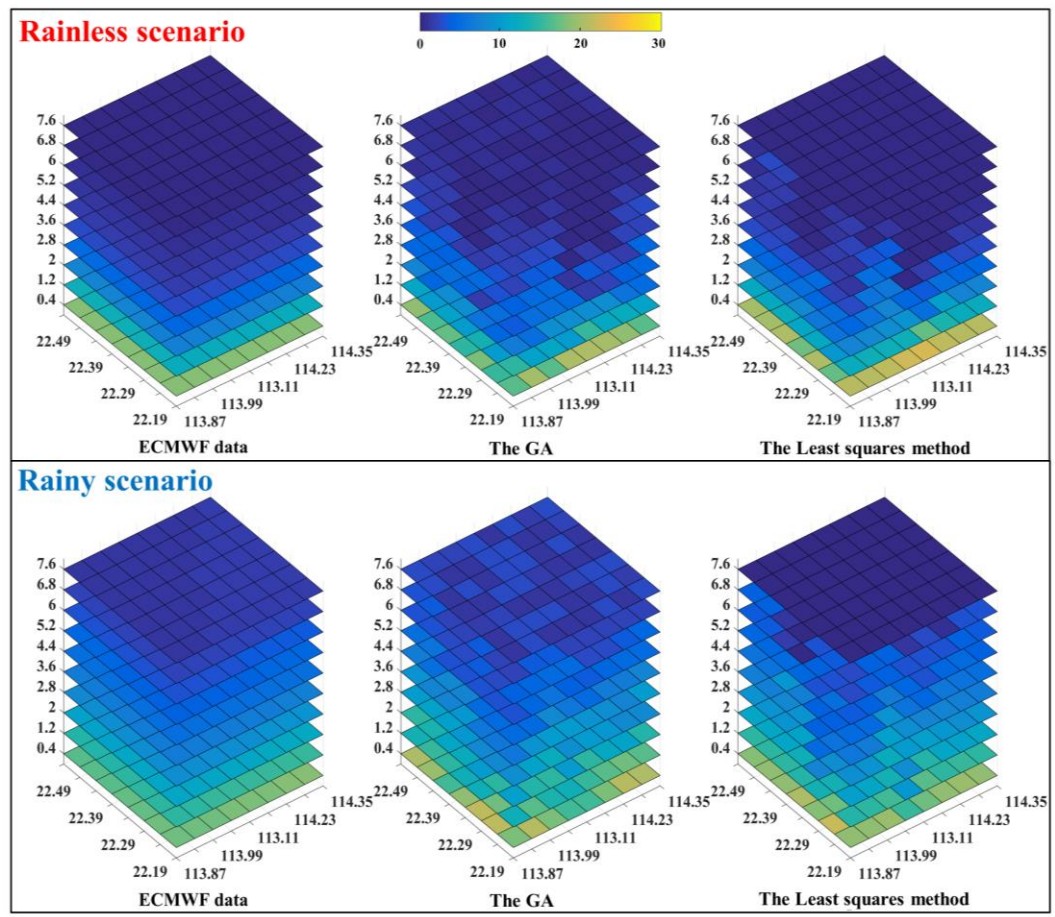

10     Fig. 12 The three-dimensional distribution of water vapor density derived from ECMWF data, the GA method and the Least squares method (upper for rainless scenario and lower for rainy scenario)



**Table 3** Statistical results of the GA and the Least squares method comparison, ECMWF data as a reference (g/m$^3$)

|  | GA method | | Least squares method | |
|---|---|---|---|---|
|  | **RMS** | **MAE** | **RMS** | **MAE** |
| **Rainy scenario** | 1.84 | 1.42 | 1.94 | 1.47 |
| **Rainless scenario** | 1.71 | 1.39 | 1.79 | 1.37 |
| **average** | 1.78 | 1.41 | 1.87 | 1.42 |

To further demonstrate the tomographic results of the GA and the least squares method, regression and boxplot are conducted and displayed in Figure 13. It covers all solutions, each of which contains 560 voxel results. In the left panel, a

5    good linear regression relationship is showed by the distribution of scatter points and the straight line of regression. Specifically, the starting point of the regression equation is 0.5198 and the slope is 0.9401. In the right panel, it shows the distribution of differences between the two types of tomographic results. The Q1 and Q3 are -0.84 and 0.60 g/m$^3$, which means more than 50% of the differences between the two methods are within 1 g/m$^3$. The upper and lower bound are 2.75 and -2.98 g/m$^3$, and outliers only account for 3.11%. Therefore, the tomographic results based on the GA has a good

10    agreement with that of least squares method in this experiment. A reliable tomographic result can be achieved by the GA without being restricted by constraint equations and matrix inversion like traditional least squares method.

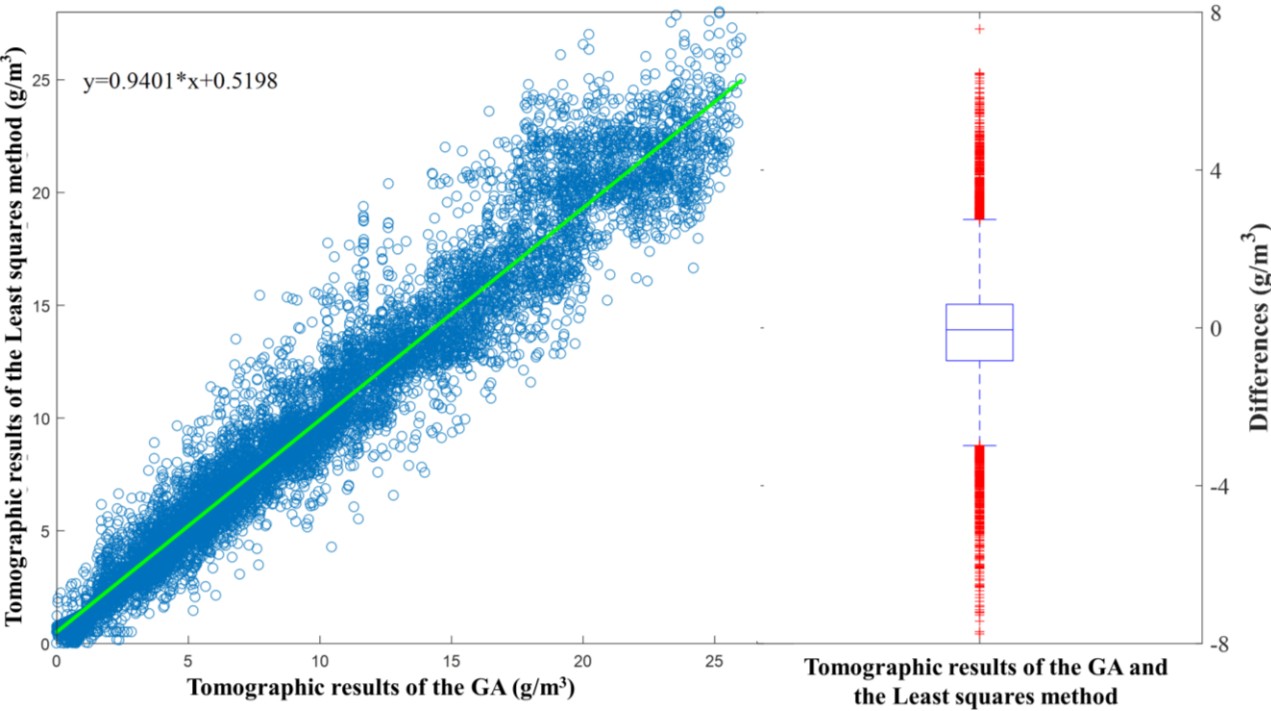

**Fig. 13** Regression (left) and boxplot (right) for tomographic results of the GA and the Least squares method



## 3.6 Analysis of results in different weather conditions

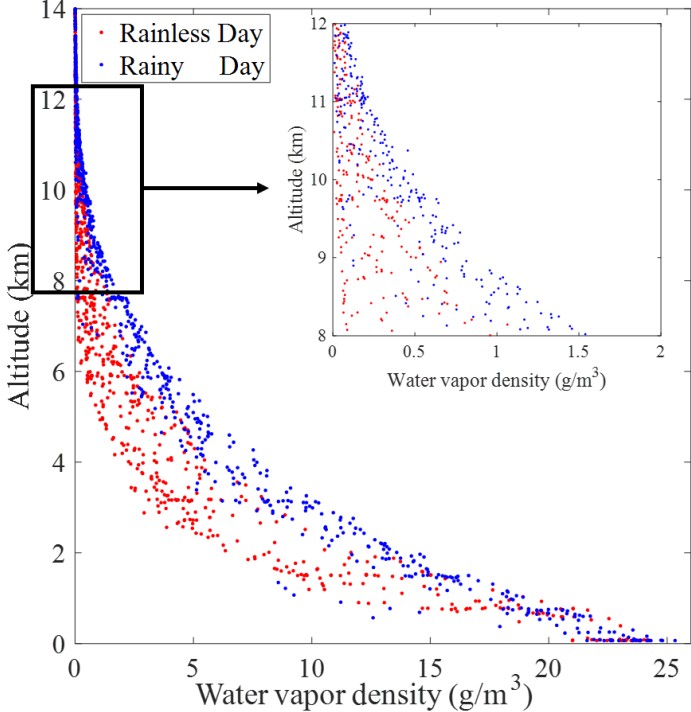

**Fig. 14** Changes of water vapor density with altitude in different weather conditions

In our experiments, the comparison under various weather conditions illustrate that tomographic result of rainless scenarios

was better than of rainy scenarios, which is the same as the conclusion of other studies (Yao et al., 2016, Zhao et al., 2017 and Ding et al., 2017). This is due to the fact that the spatial structure of atmospheric water vapor is relatively stable in rainless weather, while its spatial distribution changes faster in rainy weather. This imposes certain limitations on tomography to obtain accurate water vapor during the unstable weather conditions. Additionally, the average PWV during the two-time period were counted from the radiosonde data, 68.95 mm for rainy day and 48.26 mm for rainless day. The

more water vapor information exists in rainy weather, which may make tomography producing more errors. Moreover, all the water vapor density along the radiosonde path were collected during the experiments. Fig. 14 shows the changes of water vapor density with altitude, in which the rainy and rainless weather were represented by blue and red dots. The situation of 8-12 km is magnified to better show the water vapor information outside the tomographic region. In the figure, the larger value of WVD can be observed above 8 km in rainy days compared with that of rainless days. For the rainless situation, the

value of WVD within 8-12 km is small and near to zero, while the value is basically not close to zero in the rainy situation, especially in the range of 8-10 km, which is substantially greater than 0.5 g/m$^3$. Referring to the selection of the tomographic heights in other articles, considering the long-term statistics of water vapor in Hong Kong, and taking into account the drawbacks of the excessive number of tomographic voxels, we selected 8 km as the top boundary of the research area in this



paper, which ignores the water vapor information above 8 km in our tomographic model. Obviously, it has less influence on the accuracy of the tomographic result in rainless weather condition. For the rainy weather condition, we think the effect could be slightly larger, which is one reason why the tomographic results of rainy days were worse than those of rainless days in our experiments.

**4 Conclusions**

In this paper, a new tomography approach based on the genetic algorithm was proposed to reconstruct a three-dimensional water vapor field in Hong Kong under rainy and rainless weather conditions. The inversion problem was transformed into an optimization problem that no longer depends on excessive constraints, priori information and external data. Thus, many problems do not need to be considered, such as the difficulty of inverting the sparse matrix, the limitation and irrationality of

constraints, the weakening of tomographic technique by prior information, and the restriction of obtaining external data. Based on the fitness function established by the tomographic equation, the water vapor tomographic solution could be achieved by the genetic algorithm through the process of selection, crossover and mutation.

Our new approach is validated by tomographic experiments using GPS data collected over Hong Kong from DOY 163 to 169, 2017 (rainy days) and 225 to 231, 2017 (rainless days). The problem of matrix ill-condition was discussed and analysis

by the grayscale graph and condition number. In a comparison of the SWV residuals, internal and external accuracy testing are both used for the GA tomography. The RMS/MEA of SWV are 1.52/0.94 and 10.07/8.44 mm for the internal and external accuracy testing, respectively, which illustrates a good tomographic result. When mapping the SWV back to the zenith direction, most of the stations achieved a small RMS (<1.5 mm). It is proposed that follow-up research be undertaken to examine whether the existence of multiple stations in the same voxel will affect the tomographic medeling. The

normalization and box plot of SWV residuals were adopted in external accuracy testing, indicating a good tomographic result based on the proposed method. In addition, the water vapor density of the proposed method agreed with that of radiosonde, and the statistical results show that the RMS and MEA are 1.43 $g/m^3$ and 1.19 $g/m^3$, respectively. A close relationship between the WVD derived from tomography and that of radiosonde was also detected by linear regression analysis. The WVD residuals were displayed in the form of box plot with a small boundaries and quartiles as well as no

outliers. To better display the three-dimensional distribution of tomographic results, the ECMWF data is utilized. And Least squares method is selected as the representative of traditional tomographic method to compare with the GA, a good consistency is demonstrated in terms of RMS, MAE, linear regression and boxplot. It indicates that a reliable tomographic result can be achieved by the GA without being restricted by constraint equations and matrix inversion like traditional least squares method. Moreover, the comparison under various weather conditions illustrated that tomographic result of rainless

scenario was better than that of rainy scenario, and the reasons were discussed.




*Author contributions.* Conceptualization, Fei Yang, Jiming Guo and Junbo Shi; Data curation, Yinzhi Zhao, Lv Zhou and Di Zhang; Formal analysis, Jiming Guo and Junbo Shi; Methodology, Fei Yang; Resources, Junbo Shi, Xiaolin Meng; Validation, Fei Yang; Writing – original draft, Fei Yang; Writing – review & editing, Jiming Guo, Xiaolin Meng, Junbo Shi, Lv Zhou, Yinzhi Zhao and Di Zhang.

*Competing interests.* The authors declare no conflict of interest.

*Acknowledgement.* This research was funded by [National Natural Science Foundation of China] grant number [41604019, 41474004]. The authors would like to thank the Lands Department of HKSAR for providing the GNSS data from the HONG
KONG Satellite Positioning Reference Station Network (SatRef). Chinese Scholarship Council (CSC) and the University of Nottingham for providing the opportunity for the first author to study at the University of Nottingham for one year. Acknowledgements are also given to the editor in charge (Roeland Van Malderen) and my colleague at the University of Nottingham (Simon Roberts) for their revision to improve the English language and style of the paper.

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
