# Peer review of "A GPS water vapor tomography method based on a genetic algorithm"

_Atmospheric Measurement Techniques, 2019_

## Referee Comment (RC1) · Anonymous Referee #1 · 1 Jul 2019

The paper by Yang et al. (2019) introduces new methodological solution to the GNSS tomography ill-conditioned problem. Authors suggest to use of genetic algorithm that is applying optimization principle based on the minimization function of the slant residuals (y-Ax), and stochastic modelling of water vapour field evolution. The concept is sound, methodology quite innovative at least in the tomography community, but comparison with standard method reveals that there is a little or no improvement once the genetic approach is used. Moreover, competitive studies for the same location, shows better performance

p.16 "Xia et al. (2013) obtained a RMS of 1.01 g/m3 by adding the COSMIC profiles, Yao et al. (2016) obtained a RMS of 1.23 g/m3 by maximally using GPS observations and a RMS of 1.60 g/m3 without the operation, Zhao et al. (2017) achieved a RMS

of 1.19 g/m3 and 1.61 g/m3 considering the signal rays crossing from the side of the research area and a RMS of 1.79 g/m3 without this consideration, Ding et al. (2017) obtained a RMS of 1.23 g/m3 and 1.45 g/m3 by utilizing the new parametric methods and the traditional methods, Yao et al. (2017) achieved the RMS from 5 1.48-1.80 g/m3 using different voxel division approaches, etc, the total RMS of 1.43 g/m3 for the two time periods in this paper can be considered as a good agreement with the radiosonde data regardless of the weather conditions".

Therefore, two questions should be asked: are there any information left in the slants observations that can be utilized by the tomography framework, if positive, one might ask whether approach with introducing new algorithm to old parametrization will aid in the development of tomography processing. I suggest to address these two major questions in the revision process. Overall, the manuscript presentation quality is high, however few points need to be addressed (in addition to two major questions, stated above):

1. The genetic algorithm should be clearly explained and compared to the classic Least Square, Kalman Filter or Algebraic Reconstruction Technique solutions, reader need to understand the principles of approach and its application to the tomography problem. This comment is related to: The Introduction section where Authors only briefly p.3 l. 1-10 discuss differences between new method and standard methods, 2 Methodology where Authors should add one subsection discussing classic Least Squares applied in next section. 3. Experiment and Analysis, where reads would expect how Table 1 and steps discussed on pages 5 and 6 links to real data, It should be clear how choices of parameters from Table 1 translates into algorithm performance in more detailed, step-wise manner.

2.Comparisons with radiosondes fig 9, and with ECMWF fig 12 are corner stones of this manuscript. Therefore, it is difficult to understand why LS and Genetic algorithms were only compared to ECMWF but no to RS, as in fig 9. It should be done only for overlapping voxels. Why not to add to fig 9 two extra lines one for tomography LS and

one for ECMWF, this will clearly indicate the quality of retrieval in time.

3.The choice of research area to be one of the well-studied Hong-Kong cases has to be evaluated positively. However, division into rainy and rainless days is not supported by any meteorological analysis such as air mass origin, rain type, rain intensity, other associated phenomena. This is important as not all-weather types associated with rain will produce increase of SIWV. Moreover, there is limited evidence that the differences between so called "rainy" and "rainless" days are significant.

---

## Referee Comment (RC2) · Anonymous Referee #2 · 19 Aug 2019

General comments:

This paper proposes a new approach to solving GPS tomographic inversion to retrieve 3-D atmospheric water vapor distribution above a network of GPS receiving stations. It should circumvent the strong constraints of classical techniques that have to deal with the inversion of an often very sparse matrix. Hence, this approach could be of great interest to the community.

However, to demonstrate the good performances of their technique, the authors set to provide an ensemble of statistical indicators, some with respect to GPS slant delays, some with respect to radio-sounding profiles, some with respect to "classical techniques", etc. but usually considering only global RMS and MAE scores. Although statistical estimates are of interest, they do provide the physical/meteorological understanding of the actual objective (and challenge) of tomography inversion: it is the 3-D distribution of water vapor and more particularly its vertical variability. Hence, it would have been, in my view, much more informative and useful to the community to have comparisons of profiles of RS + ERA + GA + Least-Square + Ref GPS, possibly with the corresponding statistical metrics in order to perceive the actual capacity of the new technique to resolve the water vapor distribution, rather that producing a dispersed set of statistics which makes it difficult how the technique compares globally.

This article is globally well written and easily readable, nonetheless, the English phrasing is at times a bit awkward which might lead to some misunderstandings.

Specific comments:

P.5, L.8: after criteria, do you mean a "," or a ":"? i.e., are there 3 (",") or 2 (":") conditions for termination. In any case, a precise description of the termination criteria and how they are defined should be clearly stated here.

P.9, L.15: for clarity one could add something like "The change of tomography-computed VS GAMIT-estimated slant water vapor residuals"... Likewise, if this tells us that the GA method compares reasonably well with the original data, it would have been very interesting (and useful to evaluate the method) to know how would have fared a "classical" inversion technique.

P.9, L.17: "It is clear ... residuals decreased with ... elevation angle". Readers can read a graph. Hence, if that is stating the obvious, then the sentence can be deleted... otherwise, if that is a point of interest, than it should be discussed... Likewise with the following sentence: "The right ... angles".

P.10, L.11-13: Is there some altitude difference between the 2 stations? I would guess that if that is the case, hkmw is higher than hkpc. Actually, at this point, one could also discuss the reason why in fig 5 all zenith residuals are positive!

P.13, L.15-17: I guess this sentence relates to the green box plot of Fig. 8... but are

you sure that the range [-7.08, 4.47] is a sign of good water vapor restitution for the new method??? That is, I guess, what should be appreciated rather than a good statistical distribution!

P.14, Fig.9: it is a pity that the equivalent graphs for the no rain days are not provided. Indeed, the major limitation of GPS tomographic inversion is its ability to retrieve the vertical variability and that can only be assessed by profiles comparisons with RS, not by global statistics.

P.15, L.3-5: A this point, one could think in terms of relative error rather than absolute error.

P.18, L.7-8: If there are cases when GA performs better than lest square methods and others when it is the opposite, the authors should at least try to sort out if there are some "signature" to those contrasted behaviors (like the presence or amplitude of rain, the type of weather regimes, or more technical reasons such as GPS constellation configurations, . . . etc) in order to provide informative comments to the reader. Indeed, it is important to know how reliable the GA method is compared to the established least square ones: if it performs globally as well and is more computing effective, or performs better, than it is a real progress. If it under performs compared to others, than it has less interest.

P.19, L.10: The statement that GA can achieve good tomographic results is certainly true, but it should be discussed in light of the comment regarding the comparison with other methods (see comment above).

P.20, L.9-10: "more water vapor information exists in rainy weather"!!! That needs to be explained (or stated in an understandable way). In my view, weather conditions do not modify the amount of information but the value of such!

P.20, L.10-11: the sentence "Moreover . . . experiments" is unclear or seem unachieved. . . the reader expects something like "and . . ." to know what is the consequence of making measurements during experiments!

P.20, L.18 and following: Indeed, neglecting water vapor above 8km in near tropical conditions is far from ideal as it much below the tropopause. Hence a significant part of the water vapor distribution (and dynamics) is not considered. That actually questions the adequacy of Rain / No Rain comparisons throughout the paper at this stage as it could be explained solely by the vertical development of cloud systems.

Technical corrections:

P.1, L.16 (and throughout the text): I think one should use the term "a priori" rather than "priori" information or data.

P.1, L.28: "and are . . ." this sentence is not grammatically correct and one wonders to what this part relates to.

P.2, L.3: I guess you meant "to improve the restitution of the spatio-temporal variations".

P.8, Fig.3: Isn't there a graph issue: why is the coloring not matching the grid, for example, in the last part of the figure, there are voxels with some black and some white in it while I understand from the text that it should be either black or white only.

Conclusions:

This paper introduces a new GPS tomographic inversion method. It has true scientific potential and should make for a significant contribution to the community. Although it has all the tools to provide the necessary informative comparisons to assess the performances of the proposed method, it lacks some key analysis at this point and should be reconsidered by the authors.

In conclusion, there is currently too much "global statistics" and too little "physical/meteorological considerations" for tomographic water vapor retrieval.

---

## Author Comment (AC1) · 13 Sep 2019

**Title:** A GPS water vapor tomography method based on a genetic algorithm

**Journal:** Atmospheric measurement techniques

**Manuscript ID:** amt-2019-95

Dear Reviewer,

We would like to thank the anonymous reviewer for providing an opportunity to revise the manuscript. The comments and suggestions of the reviewer are all valuable and very helpful. We have studied them carefully and have made revisions to improve the manuscript. Revised portion are marked in **red** in the manuscript and the main corrections and additions are given below with a comment followed by a response (in red color).

Best regards,
Authors

**Comments and Suggestions for Authors**

The paper by Yang et al. (2019) introduces new methodological solution to the GNSS tomography ill-conditioned problem. Authors suggest to use of genetic algorithm that is applying optimization principle based on the minimization function of the slant residuals (y-Ax), and stochastic modelling of water vapour field evolution. The concept is sound, methodology quite innovative at least in the tomography community, but comparison with standard method reveals that there is a little or no improvement once the genetic approach is used. Moreover, competitive studies for the same location, shows better performance

p.16 "Xia et al. (2013) obtained a RMS of 1.01 g/m3 by adding the COSMIC profiles, Yao et al. (2016) obtained a RMS of 1.23 g/m3 by maximally using GPS observations and a RMS of 1.60 g/m3 without the operation, Zhao et al. (2017) achieved a RMS of 1.19 g/m3 and 1.61 g/m3 considering the signal rays crossing from the side of the research area and a RMS of 1.79 g/m3 without this consideration, Ding et al. (2017) obtained a RMS of 1.23 g/m3 and 1.45 g/m3 by utilizing the new parametric methods and the traditional methods, Yao et al. (2017) achieved the RMS from 1.48-1.80 g/m3 using different voxel division approaches, etc, the total RMS of 1.43 g/m3 for the two time periods in this paper can be considered as a good agreement with the radiosonde data regardless of the weather conditions".

  [Response]: Thank you for the comments.
In the literatures mentioned-above, the RMS achieved by Yao et al. (2016), Zhao et al. (2017) and Ding et al. (2017) are 1.60, 1.79 and 1.45 g/m$^3$, respectively, when the traditional Least squares method is used. To obtain tomographic results with higher accuracy, Xia et al. (2013) added extra COSMIC data, Yao et al. (2016) and Zhao et al. (2017) added the signals crossed through from the side faces by using Radiosonde and

ECMWF data, respectively. In the articles written by Yao et al. (2017) and Ding et al. (2017), they explored the influence of the voxel division approaches and different parametric methods on the accuracy of tomographic results. It is found that the tomographic results are stable when using the traditional Least squares method with same observation data and same grid model. The normal way to improve the accuracy is to introduce more observations, such as COSMIC data, radiosonde, ECMWF data and extra GNSS data. In our paper, the proposed GA method is conducted based on the common case, i.e. without any other extra observations. The RMS obtained in this paper is 1.43 $g/m^3$, which is not worse than the results in the literatures (1.60, 1.79 and 1.45 $g/m^3$) using the traditional Least squares method with similar conditions. Moreover, the difference of the experimental period and the grid division may affect the above comparison.

We also conducted the tomographic experiments using the traditional Least squares method in this paper. The comparison with traditional Least squares method revealed that there is a little improvement once the genetic approach is used. The focus of this paper is to solve the ill-conditioned problem of water vapor tomography using the proposed method, by which to overcome the difficulty of inverting the sparse matrix in Least squares method, the weakening of tomographic technique by a prior information in algebraic reconstruction technique and the restriction of obtaining external data. To significantly improve the accuracy of the tomographic results is not the focus of our research. Similarly, the algebraic reconstruction technique and the Kalman filter approach are also proposed to provide a new solution for water vapor tomography and to solve the shortcomings of the previous methods, rather than focusing on the significant improvement of the tomographic accuracy. In my view, the tomographic accuracy could not be significant different by different methods when the number of water vapor observations and their distribution are the same for each method. Therefore, once the COSMIC data in Xia et al. (2013) or the GPS signals crossed through by the side face in Yao et al. (2016) and Zhao et al. (2017) are introduced to the tomographic model based on the GA, we think the tomographic accuracy will be improved. But this is not the point of this paper and can be validated in the follow-up research.

Therefore, two questions should be asked: are there any information left in the slants observations that can be utilized by the tomography framework, if positive, one might ask whether approach with introducing new algorithm to old parametrization will aid in the development of tomography processing. I suggest to address these two major questions in the revision process.

[Response]: Thank you for the comments.

We think that there is no information left in the slant observations that can be utilized by the tomography framework. Since the slant observations can be used in the tomographic model are those crossed through from the top boundary. The common tomographic experiments, including our research, are modeled by this part of the slant observations. In the articles of Yao et al. (2016) and Zhao et al. (2017), they added the slant observations crossed through from the side face to the tomographic model by using the radiosonde and ECMWF data. But the accuracy of this part of the slant

observations outside the tomographic region remains to be further tested. The tomographic results in their articles showed that the accuracy is improved, it is still not the common method for water vapor tomography. The current tomographic researches are based on the slant observations with high accuracy passing through from the top boundary. Ding et al. (2017) proposed a new parametric method which use the vertex value of the voxel to represent the water vapor density of the voxel. In the common method, the value of the central point in the voxel is considered as the water vapor density of the voxel. The properties of the tomographic observation equation in the above two method are still the same. The Ding's method is not a commonly used method in water vapor tomography.

In this paper, we adopted the common method of tomographic research, which only used the slant observations crossed through from the top boundary and considered the value of the central point in the voxel as the water vapor density of the voxel, to conduct the tomography based on GA. We believe that the research based on the above method is universal and reasonable. In the follow-up research, the studies can be done by adding the slant observations passing through from the side face and using the Ding's parametric method. Actually, many similar methods were proposed to explore the improvement in tomographic accuracy, but they are not commonly used. To study the application of the genetic algorithm in water vapor tomography, it is reasonable to construct the tomographic equation by using the common methods.

Overall, the manuscript presentation quality is high, however few points need to be addressed (in addition to two major questions, stated above):

1. The genetic algorithm should be clearly explained and compared to the classic Least Square, Kalman Filter or Algebraic Reconstruction Technique solutions, reader need to understand the principles of approach and its application to the tomography problem. This comment is related to: The Introduction section where Authors only briefly p.3 l. 1-10 discuss differences between new method and standard methods, 2 Methodology where Authors should add one subsection discussing classic Least Squares applied in next section. 3. Experiment and Analysis, where reads would expect how Table 1 and steps discussed on pages 5 and 6 links to real data, it should be clear how choices of parameters from Table 1 translates into algorithm performance in more detailed, step-wise manner.

[Response]: Thank you for the suggestions.

To make this paper better understood by readers, we have fully considered the three comments and carefully revised the relevant parts of the article. According to the comments, we added appropriate information in the chapters of Introduction, Methodology, Experiment and Analysis. Below is the added content and you can see them in the revised manuscript.

"The ART techniques are iterative algorithms that proceed observation by observation. Only two vector y, x and a data structure containing the slant subpaths in each voxel are required to solve the observation equations. The algorithms consist two loops. The inner loop processes SWV by SWV and applies an adequate correction to each voxel. After

all SWVs have been executed the next iteration is started in the outer loop (Bender et al., 2011). It is not necessary to perform the matrix inversion and therefore avoids the ill-conditioned problem. But it only updates the results of the voxels traveled through by signal rays and the tomographic results heavily depend on the exact initial field, the data quality and relaxation parameter (Wang et al., 2014)."

"It assumes that the water vapor density in each voxel meet the Gauss-Markov random walk behavior for a certain period of time, and establishes the corresponding state equation of Kalman Filter. The observation vector is utilized based on the mathematical model to perform the best estimation of the state vector, which is a process of continuous prediction and correction."

"2.2 Water vapor tomography based on Least squares method
After obtaining the observation equation (Eq. (2)), three kinds of constraints are usually added:

$$0 = H \cdot x \tag{1}$$

$$0 = V \cdot x \tag{2}$$

$$0 = T \cdot x \tag{3}$$

Equations (6)-(8) are the vertical constraints, horizontal constraints and top constraints. For the horizontal constraint equation, it assumes that the distribution of water vapor density is relatively stable in the horizontal direction within a small region. Thus, the water vapor density within a certain voxel can be represented by the weighted average of its neighbors in the same layers. For the vertical constraint equation, it is a relationship established for the voxels between two adjacent layers basing on the analysis of meteorological data for many years. The top constraint is to set the water vapor density of the top boundary to a small constant. Based on the principle of Least square, the tomographic results can be achieved by the following formula:

$$x = \left( A^T A + H^T H + V^T V + T^T T \right)^{-1} \times \left( A^T y \right) \tag{4}$$

To obtain the inverse matrix in Eq. (9), the singular value decomposition is required and its detail instruction can be seen in the relevant literature (Flores et al. 2000)."

"According to the flowchart 1, the above GPS observation data were processed to construct the tomographic equation and further convert it into the fitness function for the optimization algorithm. The population size is chosen based on the total number of unknown parameters (water vapor density). The value of 200 is the default option of the algorithm when the number of unknows exceeds a certain amount. The elite count is chosen to be 10 to specifies the number of individuals that are guaranteed to survive to the next generation, since it is based on the population size (0.05 * population size). The other parameters are selected as Table 1, which are the default settings of the algorithm for the common use.
"

2. Comparisons with radiosondes fig 9, and with ECMWF fig 12 are corner stones of this manuscript. Therefore, it is difficult to understand why LS and Genetic algorithms were only compared to ECMWF but no to RS, as in fig 9. It should be done only for overlapping voxels. Why not to add to fig 9 two extra lines one for tomography LS and one for ECMWF, this will clearly indicate the quality of retrieval in time

[Response]: Thank you for the suggestion.

In the revised manuscript, we compared the GA and Least squares method using the radiosonde and ECMWF data as reference data and listed the statistical results. In the new figure (Fig. 14), we added two extra lines, one for tomography LS and one for ECMWF, to show the comparison of profiles of GA, Least squares method, radiosonde and ECMWF data during the rainless days. Fig. 9 belongs to section 3.4, the focus of which is to demonstrate the good consistency of GA tomographic results and radiosonde data.

3. The choice of research area to be one of the well-studied Hong-Kong cases has to be evaluated positively. However, division into rainy and rainless days is not supported by any meteorological analysis such as air mass origin, rain type, rain intensity, other associated phenomena. This is important as not all-weather types associated with rain will produce increase of SIWV. Moreover, there is limited evidence that the differences between so called "rainy" and "rainless" days are significant.

[Response]: Thank you for pointing it out.

We reviewed the meteorological data and provided more relevant weather information in the revised manuscript. The daily rainfall and relative humidity in different period are presented in detail. Moreover, we counted the SWV produced in the selected stations and the results in different days are listed in the table below.

Table 1. The value of SWV produced in the selected stations (unit: mm)

| DOY | 163 | 164 | 165 | 166 | 167 | 168 | 169 | Average |
|-----|-----|-----|-----|-----|-----|-----|-----|---------|
| SWV(mm) | 69.9 | 68.6 | 69.4 | 92.9 | 87.9 | 85.1 | 79.7 | 79.1 |
| DOY | 225 | 226 | 227 | 228 | 229 | 230 | 231 | Average |
| SWV(mm) | 108.5 | 109.4 | 107.8 | 108.2 | 115.3 | 123.1 | 118.4 | 112.9 |

The above listed data can show that Hong Kong experienced different weather conditions during these two periods, one with continuous rainfall and the other without rainfall. The value of SWV used for the water vapor tomography are different in the two period of time. Similar to the literature (Zhao et al. 2017, Guo et al. 2016, Yao et al, 2019), this paper is focused to prove that the water vapor tomography can achieve good results in rainy and rainless weather condition, not to show the differences between rain and rainless days are significant. However, the comparisons showed that the tomographic results in rainless day is better than those of the rainy days, which are consistent with the previous articles (Zhao et al. 2017, Guo et al. 2016, Yao et al. 2019). We tried to explain the reasons for the different tomographic results in different weather conditions at the end of the article. The research about the effects of rain type, rain

intensity and other phenomena on tomographic results is relative rare and is not the focus of current tomographic study. In the follow-up research, we would pay more attention on this issue.

---

## Author Comment (AC2) · 13 Sep 2019

**Title:** A GPS water vapor tomography method based on a genetic algorithm

**Journal:** Atmospheric measurement techniques
**Manuscript ID:** amt-2019-95

Dear Reviewer,

We would like to thank the anonymous reviewer for providing an opportunity to revise the manuscript. The comments and suggestions of the reviewer are all valuable and very helpful. We have studied them carefully and have made revisions to improve the manuscript. Revised portion are marked in **red** in the manuscript and the main corrections and additions are given below with a comment followed by a response (in red color).

Best regards,
Authors

**General Comments**

This paper proposes a new approach to solving GPS tomographic inversion to retrieve 3-D atmospheric water vapor distribution above a network of GPS receiving stations. It should circumvent the strong constraints of classical techniques that have to deal with the inversion of an often very sparse matrix. Hence, this approach could be of great interest to the community. However, to demonstrate the good performances of their technique, the authors set to provide an ensemble of statistical indicators, some with respect to GPS slant delays, some with respect to radio-sounding profiles, some with respect to "classical techniques", etc. but usually considering only global RMS and MAE scores. Although statistical estimates are of interest, they do provide the physical/meteorological understanding of the actual objective (and challenge) of tomography inversion: it is the 3-D distribution of water vapor and more particularly its vertical variability. Hence, it would have been, in my view, much more informative and useful to the community to have comparisons of profiles of RS + ERA + GA + Least-Square + Ref GPS, possibly with the corresponding statistical metrics in order to perceive the actual capacity of the new technique to resolve the water vapor distribution, rather that producing a dispersed set of statistics which makes it difficult how the technique compares globally. This article is globally well written and easily readable, nonetheless, the English phrasing is at times a bit awkward which might lead to some misunderstandings.

[Response]: Thank you for the comments.
In the community of water vapor tomography, the comparison with SWV, radiosonde and ECMWF data are commonly used to validate the tomographic results. The global RMS and MAE scores computed from the reference data (Gamit-estimated SWV, radiosonde, ECMWF data), which is adopted in almost all relevant tomographic articles,

is an effective way to evaluate the performance of tomographic method. As you said, comparisons of profiles of radiosonde, ECMWF data, GA, Least squares method could be much more informative and useful to the community. In the revised manuscript, we conducted the corresponding comparison to make up for this deficiency. We collected all the radiosonde data during the period of tomographic experiment and plotted the profiles of tomographic results and reference data. Specifically, the new figure (Fig. 14) shows the profiles of GA and Least squares method during the rainless days and utilized the radiosonde and ECMWF data as reference data.

**Specific comments:**
1. P.5, L.8: after criteria, do you mean a "," or a ":"? i.e., are there 3 (",") or 2 (":") conditions for termination. In any case, a precise description of the termination criteria and how they are defined should be clearly stated here.

[Response]: Thank you for pointing it out.

We rewritten this part in the revised version.

"The search terminates when a group of approximates meets the requirements of the fitness value. Generally, we set the stopping criteria for generation or calculation time."

2. P.9, L.15: for clarity one could add something like "The change of tomography computed VS GAMIT-estimated slant water vapor residuals". . . Likewise, if this tells us that the GA method compares reasonably well with the original data, it would have been very interesting (and useful to evaluate the method) to know how would have fared a "classical" inversion technique.

[Response]: Thank you for the suggestion.

We added the information in the revised manuscript.

3. P.9, L.17: "It is clear . . . residuals decreased with . . . elevation angle". Readers can read a graph. Hence, if that is stating the obvious, then the sentence can be deleted. . . otherwise, if that is a point of interest, than it should be discussed. . . Likewise with the following sentence: "The right . . . angles".

[Response]: Thank you for the suggestion.

We deleted the corresponding part in the revised manuscript.

4. P.10, L.11-13: Is there some altitude difference between the 2 stations? I would guess that if that is the case, hkmw is higher than hkpc. Actually, at this point, one could also discuss the reason why in fig 5 all zenith residuals are positive!

[Response]: Thank you for the comment.

The altitude of hkmw is a little higher than hkpc, their specific values are 194.95m and 18.13m, respectively. Considering that the vertical height of each layer of the tomographic model is 800m, both stations are located at the bottom of the first layer of the voxels. Thus, we think it still needs further research to discuss whether the height is the case. In fig 5, the MAE means mean absolute error, which is always a positive value. The zenith residuals of each station were calculated as the MAE that used in fig

5. P.13, L.15-17: I guess this sentence relates to the green box plot of Fig. 8. . . but are you sure that the range [-7.08, 4.47] is a sign of good water vapor restitution for the new method??? That is, I guess, what should be appreciated rather than a good statistical distribution!

[Response]: Thank you for pointing it out.

We corrected the corresponding part in the revised manuscript.

6. P.14, Fig.9: it is a pity that the equivalent graphs for the no rain days are not provided. Indeed, the major limitation of GPS tomographic inversion is its ability to retrieve the vertical variability and that can only be assessed by profiles comparisons with RS, not by global statistics.

[Response]: Thank you for the suggestion.

We added the corresponding graphs for the no rain days in the revised manuscript. In the figure (Fig. 14), the tomographic results at UTC 0:00 and 12:00 from DOY 225 to 231 derived from GA and Least squares method are compared and the radiosonde and ECMWF data are used as reference data.

7. P.15, L.3-5: At this point, one could think in terms of relative error rather than absolute error.

[Response]: Thank you for the comment.

We added the information about the relative error and rewritten the corresponding part in the revised manuscript.

"The WVD profiles reconstructed by the GA tomographic solutions are in conformity with those derived from the radiosonde data, especially in the upper troposphere from the perspective of absolute error. With respect to the relative error, the values of the voxels upper than 5km and lower than 5km are 31% and 15%, respectively. The reason for this phenomenon is that the value of water vapor in the upper layers is relatively low, even a small difference between the radiosonde and tomographic result can also lead to a large relative error, while water vapor content accounts for more than 90% below 5km near the Earth's surface."

8. P.18, L.7-8: If there are cases when GA performs better than lest square methods and others when it is the opposite, the authors should at least try to sort out if there are some "signature" to those contrasted behaviors (like the presence or amplitude of rain, the type of weather regimes, or more technical reasons such as GPS constellation configurations, . . . etc) in order to provide informative comments to the reader. Indeed, it is important to know how reliable the GA method is compared to the established least square ones: if it performs globally as well and is more computing effective, or performs better, than it is a real progress. If it under performs compared to others, than it has less interest.

[Response]: Thank you for the comment.

The focus of this paper is to solve the ill-conditioned problem of water vapor

tomography using the proposed GA method, by which to overcome the difficulty of inverting the sparse matrix in Least squares method, the weakening of tomographic technique by a prior information in algebraic reconstruction technique and the restriction of obtaining external data. To significantly improve the accuracy of the tomographic results is not the focus of our research. Similarly, the algebraic reconstruction technique and the Kalman filter approach are also proposed to provide a new solution for water vapor tomography and to solve the shortcomings of the previous methods, rather than focusing on the significant improvement of the tomographic accuracy. In my view, the tomographic accuracy could not be significant different by different methods when the number of water vapor observations and their distribution are the same for each method.

In this paper, the comparison with tomographic results of the Least squares method is to prove that the results of the GA are appreciated. Table 3 listed the numerical results including RMS and MAE during the whole experimental period and showed that the result is a little better than that of the least squares method when the ECMWF data is regarded as the true value. The solutions that least squares method yields better results than the GA does only accounts a small part. This is similar to the situation that least squares method can obtain results with different accuracy in different time period. We are concerned with the comparisons during the entire experiment, which show that the accuracy of GA is comparable to, or even a little higher than, the least squares method. The very few different solutions do not affect this conclusion. We think that it is reasonable to have these few different results. Since the proposed GA is not designed as the method to significantly improve the accuracy of the least squares method. Moreover, it does not show obvious relationship with the presence or amplitude of rain, the type of weather regimes, or GPS constellation configurations. More research is needed in the follow-up study to find the reasons. We corrected the corresponding part in the revised manuscript.

9. P.19, L.10: The statement that GA can achieve good tomographic results is certainly true, but it should be discussed in light of the comment regarding the comparison with other methods (see comment above).

[Response]: Thank you for the comment.

In the revised manuscript, a more detailed comparison between GA and Least squares method is conducted using the voxels above the radiosonde station. The changes of water vapor density derived from GA and Least squares method with altitudes in different days (rainless days) are shown in the new figure (Fig. 14), in which the radiosonde data and ECMWF data are considered as reference data. Moreover, the statistical values are computed and listed to better show the comparison of GA and Least squares method.

10. P.20, L.9-10: "more water vapor information exists in rainy weather"!!! That needs to be explained (or stated in an understandable way). In my view, weather conditions do not modify the amount of information but the value of such!

[Response]: Thank you for pointing it out.

We stated it in an understandable way in the revised version.

11. P.20, L.10-11: the sentence "Moreover . . . experiments" is unclear or seem unachieved. . . the reader expects something like "and . . ." to know what is the consequence of making measurements during experiments!

[Response]: Thank you for the suggestion.

We rewritten the corresponding part in the revised manuscript.

"Moreover, all the water vapor density along the radiosonde path were collected during the experiments and their changes with altitude were shown in Fig. 15, in which the rainy and rainless weather were represented by blue and red dots."

12. P.20, L.18 and following: Indeed, neglecting water vapor above 8km in near tropical conditions is far from ideal as it much below the tropopause. Hence a significant part of the water vapor distribution (and dynamics) is not considered. That actually questions the adequacy of Rain / No Rain comparisons throughout the paper at this stage as it could be explained solely by the vertical development of cloud systems.

[Response]: Thank you for the comment.

As you said, the tropopause is different in different region. It is important to determine the top boundary of the tomographic model. Chen and Liu (2014) said that atmospheric regions above 8.5 km should not be considered in the tomography model for Hong Kong. Otherwise, extra unknows will unnecessarily be introduced into the tomography model. Using 8.5km as the top boundary of tomographic modeling can save 43.3% of unknowns compared with using 15km. In addition, since in tomographic reconstruction only those rays entering from the top boundary of the voxel are considered, a higher top boundary implies that more rays will be rejected. Moreover, a more detailed comparison of the top boundary for the tomographic model is described in Yao and Zhao (2017). Two different height were selected as the top boundary in the paper, one is 10.4 km (Scheme 1) and the other one is 8km (Scheme 2). The results of experiment conducted once per hour show that the average utilization of signals increased by 7.51% from 51.51% (Scheme 1) to 59.02% (Scheme 2) and the percentage of voxels crossed by signals increased by 2.73%. The results of experiment conducted once per day also show a similar improvement. The average vertical water vapor profile and STD for 40 years (1974-2014) derived from a radiosonde station (45004) were collected and analyzed, which also shows that 8km is a reasonable choice of top boundary for Hong Kong tomographic model. In addition, the article entitled "Maximally using GPS observation for water vapor tomography" also discussed the choice of the top boundary for the Hong Kong tomographic model, which indicated that 8km is a good choice for Hong Kong region in term of utilization of signal rays and percentage of voxels crossed by signals. In other articles about the Hong Kong water vapor tomography (Chen and Liu, 2016; Chen and Liu, 2017; Zhao and Yao, 2018), 8km or 8.5km was selected as the top boundary, and good tomographic results were achieved.

Therefore, we think that 8km is a good choice for the top boundary in Hong Kong tomographic model considering the change of water vapor density with altitude in a long period, the utilization of signal rays and the percentage of voxels crossed by signals.

It was selected and demonstrated by previous articles.

**Technical corrections:**
1. P.1, L.16 (and throughout the text): I think one should use the term "a priori" rather than "priori" information or data.

[Response]: Thank you for pointing it out.

We corrected it in the revised manuscript.

2. P.1, L.28: "and are . . ." this sentence is not grammatically correct and one wonders to what this part relates to.

[Response]: Thank you for pointing it out.

We rewritten it in the revised version.

3. P.2, L.3: I guess you meant "to improve the restitution of the spatio-temporal variations".

[Response]: Thank you for pointing it out.

We corrected it in the revised manuscript.

4. P.8, Fig.3: Isn't there a graph issue: why is the coloring not matching the grid, for example, in the last part of the figure, there are voxels with some black and some white in it while I understand from the text that it should be either black or white only.

[Response]: Thank you for the comment.

The lower panel of each graph ((a) and (b)) is to show the distribution of voxel with (black) and without (white) sufficient signal. In our experiment, 1.79% of total SWV is taken as a criteria to distinct whether the voxel is crossed by sufficient signal or not. If the number is greater than the threshold, the color of the voxel is black, otherwise the color of the voxel is white. For example, if the number of signal rays crossing the voxel is greater than 88, the voxel is painted black in the last part of the figure (the lower panel of (b)). Thus, the there is no graph issue.

---

## Referee Report (RR1)

A GPS water vapor tomography method based on a genetic algorithm

**Review by André Sá**

I find the work presented in the paper to be interesting and worthy to be published because it presents tomographic results based on a Genetic Algorithms, which are not dependent on constraints, a priori data and external data. Additionally, because of these characteristics, a missing point that also should be highlighted is the potential interest for real-time or near-real-time analysis/applicability. However by my opinion the manuscript requires revisions to be applied before it can be accepted. Therefore I ask the authors to address the below given comments.

Comments:
It is not fully clear to me the implementation of the Genetic Algorithm. Namely the selection of the settings and the sensitivity of the algorithm concerning these settings. The setup of the stopping criteria or calculation time.

Major recommendations:

I recommend a more effective and clear description concerning my previous comment.

**P1L14-16**: "By using the proposed approach, it is not necessary to perform the matrix inversion process, and the water vapor tomography is no longer dependent on excessive constraints, a priori information and external data, which give rise to many limitations and difficulties."
I think this is a very strong sentence. The ART algorithms are iterative processes; they also do not need to perform the matrix inversion process. I use SIRT algorithms to do GNSS tomography and I do not use constraints. Please note that the tomographic solution is not tightly constraint to the a priori field. It uses the a priori field as a first guess to start the iteration, but this value can be much different from the value that the tomography will converge to in each voxel. The used term "no longer" may indicate that all the other methodologies besides the one presented here, use excessive constraints. I advice to rephrase the sentence.

P1L18-21: Please add some numerical information (values of agreement). How significant are the "high levels of agreement", some overall numbers concerning the comparative results.

P3L18-21: "The mandatory use of excessive constraints". Please read my comments in P1L14-16. It also depends on the density and configuration of the GNSS network, data quality, etc. I am able to get nice tomographic results with good convergence behavior without constraints.

---

## Editor Decision (ED1)

Dear authors,

the contents of the manuscript is now ok, but the English phrasing of newly added contributions is very poor. As you acknowledge a native English colleague for improving the English, I would ask you to lean on him again before a new version is submitted. Improving the language is not the task of the reviewers, editor and editorial office of the journal, but is the responsibility of the authors. I gave some examples in the proposed technical corrections, but these are far from being complete and a major improvement of the English, next to the ones mentioned by me, is still necessary.

Specific comments

- Page 5, line 8: which equations? Please refer to those.
- Page 7, Fig 2: a better (colour) map is required. You barely see the topography and the borders.
- Page 8: Give more details about the weather conditions at the two chosen periods. This can be short. For instance: "during the selected rainy period, the weather of Hong Kong was first affected by the approach and the passage of a severe tropical storm, named Merbok, with more than 150 mm of rainfall recorded on 13-14 June. Thereafter, from 15 and 16 June, the influence of an enhanced southwest monsoon and the development of a lingering trough of low pressure made the weather remaining unstable and rainy till 21 June. " For the rainless period, a description might be "With a ridge of high-pressure extending westwards from the Pacific to cover southeastern China on 16-18 August, a spell of fine weather prevailed ten days from 13 August to 22 August in Hong Kong."
- I would make two separate sections for Internal accuracy testing (section 3.3) and External accuracy testing (section 3.4).
- Throughout the manuscript: I recommend using the dates instead of the DOYs for the two examples, in particular in the conclusions.
- Page 11, lines 7-8: where do these ranges come from? Are these minimum/maximum values? 75/25 % quartiles? Please specify.
- Page 13, caption Fig 6: replace "in the external accuracy testing" with "for the KYC1 station, which has not been used in the tomographic modelling".
- Page 13: for the normalized RMS and normalized MAE computation, you write (on line 9) that residuals of SWV were divided by the GAMIT-estimated SWV, but did you also multiply with 100 thereafter, to obtain percentages? Please specify.
- Page 14, lines 1-3: when describing the normalized RMS and MAE, state that a consistent relative performance of the computed SWV is achieved, as they remain almost constant over all elevation angles. So, why are you adding "among all the weather conditions" here? Now it seems that different elevation angles reflect different weather conditions, which is obviously not necessarily true.
- Page 13 and 14, captions figures 7 and 8: explain which SWV residuals are referred too (you might also refer to Fig. 6). "SWV residuals" is too general and therefore meaningless.
- Page 17:. When comparing the RMS between the GA tomography and the radiosonde WVD profiles with RMS from other studies, are referring to studies over Hong Kong as well? And which references have those studies used to calculate the RMS with?

Radiosondes as well? Please specify! Also mention if those methods are using the more common approach for tomography (least squares) and which constraints or external datasets have been used. Start this discussion with "We compare those values with the results obtained from other Hong Kong tomographic experiments. For example, …" instead of the current awkward formulation.

- Page 19, Fig 12. Please give titles to the axis (altitude, latitude, longitude) and give the units of the colour scale.
- Page 20, Fig. 13: mention in the figure caption which "results" you are comparing (WVD).
- Page 22, Fig 15 caption: specify which dataset (radiosondes) is used to construct this figure and which periods (the usual one, I assume, but it is always good to note, as some readers primarily look at the figures when going through a paper).
- Page 23: line 3: Explain why larger values of slant water vapour lead to more errors in the tomography approach. And do higher PWV values (slant water vapour in zenithal direction) always lead to higher slant water vapor values, as you assume here?
- Page 23-24: The conclusions need to be rewritten. Instead of presenting the most important conclusions, you give a various location just a summary of the analyses you have been doing (e.g. "The WVD residuals were displayed in the form of box plot with a small boundaries and quartiles as well as no outliers").Additionally, first of all, mention the dates for the two periods, not the DOYs. Secondly, you mentioned several times "internal/external accuracy testing", but these general concepts should be more specified. Now, these concepts are rather meaningless. Describe very shortly the comparisons you did for internal/external accuracy testing when given the numbers etc. Thirdly, instead of writing "the statistical results show that", just give immediately the statistical parameters you have been considering (RMS and MEA): every reader knows that these are statistical parameters.

Technical corrections

- Page 1, line 11: Replace "part in" with "substituent of"
- Page 1, line 13: replace "as a research point in the fields of" with "as a research area in the field of"
- Page 1, lines 14-15: replace "dose" with "does"
- Page 1, line 16: remove "which give rise to many limitations and difficulties"
- Page 1, line 16-18: replace with "Experiments in Hong Kong under rainy and rainless conditions using this approach show that …"
- Page 2, line 1: replace with "Thanks to the development of GPS station networks providing atmospheric information under all weather conditions, GPS is considered …"
- Page 3, line 1: give references of the MART and SIRT techniques and remove the brackets before e.g. and after (SIRT).
- Page 3, line 2: Only two vectorS
- Page 3, line 11: replace with "The used observation vector is based on…"
- Page 3, line 19: replace "usually" with "commonly"
- Page 3, lines 21-22: Replace with "External data cannot be used in all tomographic experiments."

- Page 4, lines 1-4: be consistent and use commas (,) in the summation, not a mixture of , ; .
- Page 4: Drop "and" before SWD.
- Page 4, line13: Replace with "The ZTD is the primary parameter retrieved with GPS and is a spatially averaged parameter. If pressure measurements are available…"
- Page 4, line 17: replace ; with ,
- Page 4, line 23: replace with "The horizontal constraint equation assumes …"
- Page 4, line 25: replace with "of its neighbouring voxels in the same layers".
- Page 4, line 26: replace with "The vertical constraint equation is a relationship … layers based on …"
- Page 4, line 27: replace with: "The top constraint is obtained by setting the water …"
- Page 5, lines 4-5: replace with: … is required. More details on this technique can be found in e.g. Flores et al. (2000).
- Page 5, line 18: … that best fitS"
- Page 6, lines 8-9: replace with "Based on these steps, the optimal solution of Eq. (10) is derived, ….
- Page 6, Fig. 1: CoEfficient Matrix
- Page 8, line 6: replace "were ready for" with "are available in"
- Page 8, line 15: replace "It is defined as" with "This period represents"
- Page 8, lines 31-32: Replace with "Fig. 3 illustrates this in the form of a grayscale graph for two different days : DOY 225 at UTC 00:00 (give the date and year here instead of the DOY), a rainless day (a), and DOY 164 (give the date and year here instead of the DOY) at UTC 12:00, a rainy day (b).
- Page 9, lines 8-9: Replace with "For the two examples shown, the number of total SWV, …"
- Page 11, lines 9-10: Replace with "A particular outlier is the HKMW station, with RMS and MAE values 1.81/1.53 and …"
- Page 11, line 13: replace "This specific impact should be discussed …" with "This hypothesis will be further investigated in future research".
- Page 11: drop the last sentence of this page or explain on which ground you come to this statement.
- Page 12, caption Fig. 5: circleS, diamondS
- Page 12, line 6: replace with "for those different weather conditions"
- Page 12, line 7, drop "and the reds are generally smaller than the blues, whether in the situation of MEA or RMS". This is such an non-scientific statement!!!
- Page 12, lines 6-8: Reformulate into "From this figure, it can be noted that all MAE and RMS are below 15 mm, with average values lower for rainless days than for rainy days, respectively 8.75/7.33 and 11.38/9.54 mm. for RMS/MAE.
- Page 13, caption Fig 7: "for each elevation bin"
- Page 13, line 6: "slant water vapour outputs"
- Page 14, line 1: replace with "In terms of normalized RMS and MAE, we note that they remain …"
- Page 14, line 6: replace "demonstrates" with "points to"
- Page 14, line 11-12: Replace with "IQR is the interquartile range, defined as the difference …"
- Page 15, line 6. Replace with "As the radiosondes are launched daily at 00:00 and 12:00 UTC, the … at these times were compared".

- Page 16, line 1. Replace with "It is clear from the profiles that the WVD decreases with increasing height."
- Page 16, lines3-4. Replace with "in the upper troposphere in absolute terms".
- Page 16, line 7: replace with "while the water vapor content resides for more than 90% below …"
- Page 16, caption Table 2, and line directly below table 2: replace with "RMS and MAE of the water vapor density …"
- Page 17, line8: please be more specific on "utilizing the new parametric methods and the traditional methods": which methods are you referring to?
- Page 18, lines 6-7: replace with "vapor density can be achieved with high accuracy by tomography based on the GA. The corresponding box plots are shown in Fig. 11. It can be noted that the WVD residuals …"
- Page 18, Fig. 11: Rainy dayS, Rainless dayS. And in the caption of this Figure: specify that the WVD residuals are computed between GA tomographic approach and radiosondes.
- Page 18, line 18: Replace with "… can be found in detail in e.g. Flores …"
- Page 19, line 1: Replace "according to the pressure layer" with "at different pressure levels".
- Page 19, line 2: Replace "can obtain" with "give"
- Page 19, line 3: Drop "For tomographic results".
- Page 20, lines 1-2: Replace with "To further analyze the tomographic results of the GA and the least squares method, regression and boxplot are constructed and displayed in Fig. 13, which covers all solutions, each of them containing 560 voxel results."
- Page 20, line 4: Change to "The right panel shows"
- Page 20, line 6: add "that" after means
- Page 20, line7: replace with Consequently, the tomographic results based on the GA are in agreement with those of the least squares method for this experiment".
- Page 21, line 4: Remove "The numerical results including". This is such an awkward and meaningless expression.
- Page 21, line 7: Replace "a little" with "slightly".
- Page 22, line 7: Replace with "which is also concluded in other studies (Yao et al. …)".
- Page 23, line 13: "we think" is not a very scientific statement. You can quantify the relative amount of WVD that you are missing in your tomography experiment by neglecting the layers above 8 km.

---

## Author Response (AR2)

**Title:** A GPS water vapor tomography method based on a genetic algorithm

**Journal:** Atmospheric measurement techniques
**Manuscript ID:** amt-2019-95

**Comments to the Author:**
Dear colleague,

please take into account the remaining remarks by the two reviewers. Important remaining points here are a more in-debt classification between rainy and rainless weather, based on the references given by the reviewer #1. Referee #3 is not entirely satisfied with the description of the implementation of the algorithm on the selection of the parameters settings and the sensitivity of the algorithm concerning these settings (like the setup of the stopping criteria or calculation time as examples). Please be also more quantitative P1L18-21. Finally, to show the value of your research, please also describe its potential interest for real-time or near-real-time analysis or applications.

With kind regards,
Roeland Van Malderen

**Reply to the editor:**
Dear Roeland,

We would like to thank you and the reviewers for providing an opportunity to revise the manuscript. The comments and suggestions of the reviewers are all valuable and very helpful. We have studied them carefully and have made revisions to improve the manuscript. Revised portions are marked in yellow in the new version and a point-to-point reply to the comments is also provided.
I have checked the weather of June and August 2017 in Hong Kong through the websites(http://www.hko.gov.hk/wxinfo/pastwx/mws2017/mws201706.htm and http://www.hko.gov.hk/wxinfo/pastwx/mws2017/mws201708.htm) to verify the weather type. In the revised version, I further clearly explained the meaning of the rainy and rainless days in our paper. A more detailed reply to Reviewer #1 can be found in the corresponding response below.
To resolve the comments of referee #3, more information about the implementation of Genetic Algorithm was added to make the corresponding part clearer for the readers. The selection of the settings, the sensitivity of the algorithm concerning these settings and the setup of the stopping criteria have been described. The sentences mentioned by the referee have been rephrased in the revised version. Moreover, we have highlighted the potential interest for real-time or near-real-time analysis/applicability.

Best regards,
Authors

**Referee #1: Suggestions for revision**

I strongly suggest Authors to avoid simplification into "rainy" and "rainless" atmosphere as there might be quite different regimes guiding flows and vertical and horizontal variability involved. I suggest to verify the weather type called as a "rainy" by checking http://www.hko.gov.hk/wxinfo/pastwx/mws2017/mws201706.htm and following one of the classical review papers for instance:

Houze Jr, R. A. (1977). Structure and dynamics of a tropical squall–line system. Monthly Weather Review, 105(12), 1540-1567.

Houze Jr, R. A. (1982). Cloud clusters and large-scale vertical motions in the tropics. Journal of the Meteorological Society of Japan, 60(1).

Houze Jr, R. A. (2010). Clouds in tropical cyclones. Monthly Weather Review, 138(2), 293-344.

Otherwise the paper is well written.

[Response]: Thank you for your comments.

I have checked the weather of June and August 2017 in Hong Kong through the websites(http://www.hko.gov.hk/wxinfo/pastwx/mws2017/mws201706.htm and http://www.hko.gov.hk/wxinfo/pastwx/mws2017/mws201708.htm) to verify the weather type.

On 11 June, an area of low pressure over the South China Sea developed into a tropical storm, named Merbok. On 12 June Merbok moved across the northern part of the South China Sea and intensified further into a severe tropical storm that night. It traversed the eastern part of Hong Kong waters and made landfall over the Dapeng Peninsula before midnight. Later on 12 June, local winds strengthened significantly with heave squally showers in Hong Kong with the approach of Merbok. As Merbok weakened over land, its rainbands continued to affect Hong Kong with gusty winds and heavy rain. On 13-14 June, more than 150 millimeters of rainfall were generally recorder, with rainfall over the urban areas exceeding 250 millimeters. On 15 and 16 June, the influence of an enhanced southwest monsoon and the development of a lingering trough of low pressure made the weather remained unstable and rainy till 21 June.

With a ridge of high-pressure extending westwards from the Pacific to cover southeastern China on 16-18 August, a spell of fine weather prevailed ten days from 13 August to 22 August in Hong Kong.

From the information mentioned-above, we can see that Hong Kong suffered heavy rain from 12 June to 18 June and fine weather from 13 August to 19 August, respectively. Moreover, we provided information about relative humidity and SWV produced in the selected stations in these two periods. The daily rainfall is 0mm, the relative humidity is 75% in average and the average SWV produced in the selected stations is 79.1 mm in the period from 13 August to 19 August. For the other period from 12 June to 18 June, the maximum daily rainfall is up to 203.7 mm and the daily rainfall is 66.8 mm in average, the relative humidity and the SWV produced in the selected stations are 89% and 112.9 mm.

Therefore, we define the period from 12 June to 18 June as the rainy period, which means that continuous rainfall occurs, the relative humidity is large and the larger SWV is generated. For the rainless period, it means that fine weather occurs, the relative humidity is small and the smaller SWV is generated. In the revised version, we further clearly explained the meaning of the rainy and rainless days in our paper.

In addition, we can see the definition of weather types in other related articles about water vapor tomography. For example, Zhao et al. (2017) wrote that "DOY 87 was a sunny day and DOY 89 was a

rainy day with a total precipitation of 115.6mm according to the records of the Hong Kong Observatory" written by Zhao et al. (2017), Zhang et al. (2017) wrote that "One is from 20 July to 26 July when heavy rain attacked Hong Kong with the largest daily rainfall (~190mm) in 2015 on 22 July. The other is from 1 August to 7 August when the weather is rainless", Ding et al. (2017) mentioned that
5    "Thunderstorms continued to affect the experimental region during the rainy days from 9 to 15 August 2015. During the rainy days, the moisture content increased and changed dramatically in the troposphere. In the dry weather from 2 to 8 August, conditions became very hot with plenty of sunshine and maximum temperatures exceeded 33°C", Yao et al. (2019) also divided the weather type during experimental period into rainy and non-rainy scenario based on the rainfall information from the Hong
10   Kong Observatory. I really understand that there might be quite different regimes guiding flows and vertical and horizontal variability involved. However, it is not the focus of current tomographic study. In the relevant articles, water vapor tomography experiments were carried out over a period of time. They classified the experiments as rainy scenario and rainless scenario based on the weather condition, such as the rainfall information. The adoption of rainy and rainless days for the weather type is to verify the
15   availability and performance of the tomography in the case of significantly different weather conditions, such as sunny days without any rainfall and days with continuous heavy rainfall.

**Referee #3: Suggestions for revision**
20   Review by André Sá
I find the work presented in the paper to be interesting and worthy to be published because it presents tomographic results based on a Genetic Algorithms, which are not dependent on constraints, a priori data and external data. Additionally, because of these characteristics, a missing point that also should be highlighted is the potential interest for real-time or near-real-time analysis/applicability. However, by
25   my opinion the manuscript requires revisions to be applied before it can be accepted. Therefore, I ask the authors to address the below given comments.
[Response]: Thank you for the suggestion.
We have highlighted the potential interest for real-time or near-real-time analysis/applicability in the revised version.

It is not fully clear to me the implementation of the Genetic Algorithm. Namely the selection of the settings and the sensitivity of the algorithm concerning these settings. The setup of the stopping criteria or calculation time.
Major recommendations:
35   I recommend a more effective and clear description concerning my previous comment.
[Response]: Thank you for the comments.
In the revised manuscript, more information about the implementation of Genetic Algorithm was added to make the corresponding part clearer for the readers. The selection of the settings, the sensitivity of the algorithm concerning these settings and the setup of the stopping criteria have been described.
40

P1L14-16: "By using the proposed approach, it is not necessary to perform the matrix inversion process, and the water vapor tomography is no longer dependent on excessive constraints, a priori information

and external data, which give rise to many limitations and difficulties." I think this is a very strong sentence. The ART algorithms are iterative processes; they also do not need to perform the matrix inversion process. I use SIRT algorithms to do GNSS tomography and I do not use constraints. Please note that the tomographic solution is not tightly constraint to the a priori field. It uses the a priori field as a first guess to start the iteration, but this value can be much different from the value that the tomography will converge to in each voxel. The used term "no longer" may indicate that all the other methodologies besides the one presented here, use excessive constraints. I advise to rephrase the sentence.

[Response]: Thank you for the suggestion.

I have rephrased the sentence in the revised manuscript.

P1L18-21: Please add some numerical information (values of agreement). How significant are the "high levels of agreement", some overall numbers concerning the comparative results.

[Response]: Thank you for the suggestion.

I have added the numerical information in the revised version.

P3L18-21: "The mandatory use of excessive constraints". Please read my comments in P1L14-16. It also depends on the density and configuration of the GNSS network, data quality, etc. I am able to get nice tomographic results with good convergence behavior without constraints.

[Response]: Thank you for pointing it.

I have rewritten the corresponding part in the revised version.

[revised manuscript text omitted]

10  This period is defined as rainless days, which means that fine weather occurs without any rainfall, the relative humidity is small and the smaller SWV is generated. The other period is from 12 June, 2017 to 18 June, 2017 (DOY of 163 to 169, 2017). In this period of time, the maximum daily rainfall is up to 203.7 mm and the daily rainfall is 66.8 mm in average. For the relative humidity and the SWV produced in the selected stations, the average values are 89% and 112.9 mm, respectively. It is defined as rainy days, indicating that continuous rainfall occurs, the relative humidity is large and the larger SWV is

15  generated. The period covered is 0.5h for each tomographic solution. The radiosonde data, collected twice daily at 00:00 and 12:00 UTC in these two periods, were treated as the reference data.

According to the flowchart represented in Fig. 1, the above GPS observation data were processed to construct the tomographic equation and further convert it into the fitness function for the optimization algorithm. The population size is chosen based on the total number of unknown parameters (water vapor density). The value of 200 is the default option of the

20  algorithm when the number of unknows exceeds a certain amount. The reproduction of elite count is chosen to be 10 to specify the number of individuals that are guaranteed to survive to the next generation, since it is based on the population size (0.05 * population size). The crossover fraction is set to the default value of 0.8 to specify the fraction of the next generation that crossover produces. In this study, generation is chosen as the stopping criteria and 100*Number of Variables is the default. The other parameters including Roulette, Intermediate and Adaptive Feasibility are selected since they are the

25  most commonly used settings for genetic algorithms. Some other selection function as well as crossover function and mutation function can be adopted in the genetic algorithm, and the 
[revised manuscript text omitted]

---

## Author Response (AR3)

**RESPONSE TO COMMENTS**

**Title:** A GPS water vapor tomography method based on a genetic algorithm

**Manuscript ID:** amt-2019-95

Dear Editor,

We have corrected the language errors you listed and gone through the manuscript. The English language and style of the paper have been revised. In addition, the comments and suggestions are all valuable and very helpful. The revised portions are marked in red in the manuscript and the response are given below (in red color).

Best regards,
Authors

1. Page 5, line 8: which equations? Please refer to those.
[Response]: It refers to Eq. (2). We have corrected it in the revised version.

2. Page 7, Fig 2: a better (colour) map is required. You barely see the topography and the borders.
[Response]: We provided a new map in the revised version.

3. Page 8: Give more details about the weather conditions at the two chosen periods. This can be short.
[Response]: We have rephrased the corresponding part in the revised version.

4. I would make two separate sections for Internal accuracy testing (section 3.3) and External accuracy testing (section 3.4).
[Response]: Thank you for the suggestion. We have made the separation in the revised version.

5. Throughout the manuscript: I recommend using the dates instead of the DOYs for the two examples, in particular in the conclusions.
[Response]: We have replaced the DOYs with the dates in the revised version.

6. Page 11, lines 7-8: where do these ranges come from? Are these minimum/maximum values? 75/25 % quartiles? Please specify.
[Response]: These are minimum/maximum values, we have specified it in the revised version.

7. Page 13, caption Fig 6: replace "in the external accuracy testing" with "for the KYC1 station, which has not been used in the tomographic modelling".
[Response]: Thank you for the suggestion. We have corrected it.

8. Page 13: for the normalized RMS and normalized MAE computation, you write (on line 9) that residuals of SWV were divided by the GAMIT-estimated SWV, but did you

also multiply with 100 thereafter, to obtain percentages? Please specify.

[Response]: It was multiplied with 100 to obtain percentage. We have specified it in the revised version.

9. Page 14, lines 1-3: when describing the normalized RMS and MAE, state that a consistent relative performance of the computed SWV is achieved, as they remain almost constant over all elevation angles. So, why are you adding "among all the weather conditions" here? Now it seems that different elevation angles reflect different weather conditions, which is obviously not necessarily true.

[Response]: Thank you for the comment. We have rephrased the sentence in the revised version.

10. Page 13 and 14, captions figures 7 and 8: explain which SWV residuals are referred too (you might also refer to Fig. 6). "SWV residuals" is too general and therefore meaningless.

[Response]: The SWV residuals means the differences between the tomography-computed SWV and the GAMIT-estimated SWV. We have explained it in the revised version.

11. Page 17:. When comparing the RMS between the GA tomography and the radiosonde WVD profiles with RMS from other studies, are referring to studies over Hong Kong as well? And which references have those studies used to calculate the RMS with? Radiosondes as well? Please specify! Also mention if those methods are using the more common approach for tomography (least squares) and which constraints or external datasets have been used. Start this discussion with "We compare those values with the results obtained from other Hong Kong tomographic experiments. For example, …" instead of the current awkward formulation.

[Response]: Thank you for your comment. We have rephrased the corresponding parts in the revised version.

12. Page 19, Fig 12. Please give titles to the axis (altitude, latitude, longitude) and give the units of the colour scale.

[Response]: We have provided a new figure containing the titles to the axis and the units of the color scale in the revised version.

13. Page 20, Fig. 13: mention in the figure caption which "results" you are comparing (WVD).

[Response]: We have mentioned it in the revised version.

14. Page 22, Fig 15 caption: specify which dataset (radiosondes) is used to construct this figure and which periods (the usual one, I assume, but it is always good to note, as some readers primarily look at the figures when going through a paper).

[Response]: We have specified them in the revised version.

15. Page 23: line 3: Explain why larger values of slant water vapour lead to more errors in the tomography approach. And do higher PWV values (slant water vapour in zenithal direction)

always lead to higher slant water vapor values, as you assume here?

[Response]: Thank you for pointing it out. We found that it is not a cause of the different tomographic results on rainy and rainless days. We have deleted the corresponding parts in the revised version.

16. Page 23-24: The conclusions need to be rewritten. Instead of presenting the most important conclusions, you give a various location just a summary of the analyses you have been doing (e.g. "The WVD residuals were displayed in the form of box plot with a small boundaries and quartiles as well as no outliers").Additionally, first of all, mention the dates for the two periods, not the DOYs. Secondly, you mentioned several times "internal/external accuracy testing", but these general concepts should be more specified. Now, these concepts are rather meaningless. Describe very shortly the comparisons you did for internal/external accuracy testing when given the numbers etc. Thirdly, instead of writing "the statistical results show that", just give immediately the statistical parameters you have been considering (RMS and MEA): every reader knows that these are statistical parameters.

[Response]: Thank you for the comments. We have rewritten the corresponding parts in the revised version.

[revised manuscript text omitted]